# Green production efficiency of China's hog breeding industry: Spatial divergence and its driving factors

**Yifan Ji[1], Zejun He[2], Ningjie Li[1], Chun Li[1], Tao Xu[1] ***

**1** International Business School, Hainan University, Haikou, Hainan, China, **2** College of Economic and Management, Henan Agricultural University, Zhengzhou, Henan, China

* xutao_2013@outlook.com

**Data Availability Statement:** The data underlying the results presented in the study are available from EPSDATA. All relevant data are within the paper and its Supporting Information files.

## Abstract

This paper analyzes the green production efficiency (GPE) and spatial divergence of the hog breeding industry, with the aim of providing a foundation for the rational layout of hog breeding and promoting the industry's high-quality development. The paper uses the SBM model to estimate GPE in 29 provinces, cities, and districts of China from 2006 to 2019. Furthermore, it analyzes the spatial divergence of GPE and its driving factors using divergence indexes and the Geodetector. The results show that China's GPE of the hog breeding industry increased from 0.409 to 0.496 between 2006 and 2019. The highest efficiency occurred during the I-period, while the lowest efficiency was observed during the II-period. The highest efficiency was in the key development region, and the lowest efficiency was in the potential growth region. The spatial divergence of GPE in the hog breeding industry expanded, with labor input, non-point source pollution, resource endowment, and environmental load bearing being the main driving factors for the expansion in each period. The potential growth region had the largest spatial divergence, mainly affected by resource endowment. In contrast, the constrained development region had the smallest spatial divergence, mainly affected by resource endowment and pollutant emissions. The spatial divergence of moderate and key development regions was considerable, mainly affected by environmental investment, environmental load bearing, and pollutant emissions. Therefore, the key to improving the GPE of the hog breeding industry is to promote the adoption of advanced technology, such as labor-saving, material-saving, and emission reduction technologies. Moreover, several actions should be taken to reduce the spatial divergence among different regions, such as integrated breeding, clean standards, large-scale breeding, and high-end boutique.

## Introduction

The hog breeding industry plays a crucial role as a core sector and a major component of the livestock industry worldwide. Pork production accounts for 32.57% of the total meat output globally Data source: EPS data platform (https://www.epsnet.cn) for 2020 data., with

**Funding:** This study was supported by the National Natural Science Foundation of China [grant numbers 72003054]; the Major Consulting Projects of Chinese Academy of Engineering [grant numbers 2020-XZ-19]; the Science and Technology Think Tank Research Project of Henan [grant numbers HNKJZK-2021-31C]; and Youth Project of Natural Science Foundation in Hainan [grant number 720QN243].

**Competing interests:** On behalf of all authors, disclose any competing interests that could be perceived to bias this work—acknowledging all financial support and any other relevant financial or non-financial competing interests. This does not alter our adherence to PLOS ONE policies on sharing data and materials.

100.853 million tons of pork consumed worldwide in 2021 Data source: USDA Data (https://www.usda.gov/topics/data)., serving as a significant protein source for human consumption. Nevertheless, hog breeding is a highly polluting industry that faces prominent resource and environmental constraints [1]. On the one hand, as the cost of land, labor, and energy continues to increase, the resource dividend is gradually disappearing. On the other hand, the heavy and difficult-to-treat manure emissions have adverse effects on the ecological environment [2]. Particularly, China, as the largest pork consumer and producer globally, where pork consumption accounts for 44% of the world's total consumption and production accounts for 38.3% of the total global production Data source: EPS data platform (https://www.epsnet.com.cn) for 2020 data., continues to face increasingly serious environmental challenges while improving pork production efficiency. It is a long way to go to achieve China's double carbon goal. Therefore, China's hog breeding industry has reached a crucial stage of high-quality development that demands equal emphasis on both "efficiency" and "decontamination" [3]. The practice of high-quality development in the Chinese hog breeding industry can significantly contribute to the global hog industry's development and environmental improvement.

As a crucial metric for evaluating industrial efficiency and competitiveness, Green Production Efficiency (GPE) serves as a vital benchmark for gauging the level of high-quality development achieved by the hog breeding industry. GPE is a study of the degree of coordination between economic development and environmental resources, based on the limited environmental carrying capacity, to achieve the goal of energy conservation, consumption reduction, and pollution reduction in the production process, using advanced management methods and technology. It not only evaluates the efficiency of resource inputs, such as land, labor, and capital but also takes environmental factors into account to comprehensively measure the overall efficiency of industries under the constraints of resource and environmental factors [4]. GPE has emerged as an important reference for assessing the status of high-quality industrial development and providing theoretical support and policy guidance for subsequent development [5].

Scholars have shown great interest in GPE, conducting extensive research on its measurement and application, focusing on its levels in regions, cities, enterprises, and industries, as well as various key factors that affect it [6–8]. In terms of specific studies at the industry level, researchers have concentrated on the GPE of highly polluting and socially focused industries such as industry, manufacturing, and agriculture (plantation). However, studies on GPE in the livestock industry are relatively scarce. Only Han et al. [9] measured the environmental TFP of the livestock industry in each province of China and applied spatial econometric models to explore its influencing factors. Zhong et al. [10, 11] measured the GTFP of hens and dairy cows in different scales and regions in China. Research on GPE of hog breeding industry is even rarer. Zhao et al. [12] evaluated the GPE of hog breeding of various scales in 18 provinces of China based on the calculation of pollutant emission, Zhong et al. [13] conducted a detailed analysis of the efficiency of different scale breeding based on the comprehensive evaluation of hog GPE. Although numerous studies have been conducted on GPE in various sectors, the hog breeding industry's functional areas still lack sufficient research attention, failing to account for the unique GPE characteristics that differ across these areas. Conducting more targeted and detailed research on GPE in the hog breeding industry is crucial to achieving high-quality development in this field.

Due to variations in resource endowments, natural conditions, and economic development levels, the GPE of each functional area in hog breeding differs and exhibits significant spatial divergence. This spatial divergence not only leads to an unbalanced allocation of resources and technologies and a loss of coordination in industrial development, but also affects the effectiveness of emission reduction and environmental management, resulting in social green inequity and hindering high-quality development in the industry [14]. Studies have extensively

analyzed the current situation of GPE spatial divergence and evolution at the overall agricultural level, focusing on GPE index divergence and evolutionary trends, and have confirmed the temporal, spatial, and regional variability and clustering characteristics of agricultural GPE changes [5, 15, 16]. However, agricultural GPE is significantly different due to varying development levels and positioning in different economic regions, and shows spatial dependence among regions [17, 18]. Researchers have explored the causes of GPE in each region, focusing on the effects of economic factors such as the level of economic development, agricultural trade, agro-industrial agglomeration, agricultural output growth, and scale level on GPE [19–21]. With the progress of industry and the development of the green concept, attention has gradually shifted to the influence of environmental factors on production efficiency [22, 23]. Existing studies have confirmed the existence of spatial differentiation in agricultural GPE, but the extent of spatial divergence of livestock GPE, such as hog breeding, has received less attention, and insufficient focus has been given to the sources of divergence. Moreover, while the causes of GPE levels and evolution have received sufficient attention, few studies have explored the drivers of GPE spatial divergence from both internal and external perspectives, leading to a lack of basis for coordinated industrial development strategies.

Concluding this paper, statistical data is employed to measure the GPE of the hog breeding industry in 29 provinces of China from 2006 to 2019 based on the SBM model. Additionally, divergence indexes and Geodetector are utilized to explore the spatial divergence characteristics and its driving factors, respectively. Compared to previous research, this paper's marginal contributions are as follows: firstly, it concentrates on the spatial divergence of GPE in the hog breeding industry and scrutinizes the sources of its spatial divergence, providing a reference basis for the industry's green and coordinated development. Secondly, it examines the GPE characteristics in different areas based on the division of functional areas of the hog breeding industry to enhance the precision and relevance of research outcomes. Thirdly, the paper analyses the driving factors of spatial divergence of GPE in hog breeding industry from both internal and external perspectives, exploring the causes of its spatial divergence. The study results can serve as a reference for promoting coordinated regional development of the hog breeding industry, and thus improving the quality of its development.

## Methods and materials

This chapter presents a comprehensive account of the methods and data sources employed in the research process of this paper. It elucidates the procedure for choosing indicators to measure GPE in the hog breeding industry, along with the underlying rationale for selecting the drivers. This chapter furnishes methods and material backing for calculating and analyzing the research findings in the subsequent section.

### Data

This study considers 29 provinces, cities, and districts as the DMU for measuring the industrial development pattern influenced by China's development plan. To ensure comprehensive coverage, the five provinces and districts of Hong Kong, Macao, Taiwan, Tibet, and Ningxia were excluded due to missing data. The sample period is divided into three periods: the I-period (2006~2010), the II-period (2011~2015), and the III-period (2016~2019) Classification basis: the *Eleventh Five-Year Plan for the Development of National Animal Husbandry (2006–2010)*, the *Twelfth Five-Year Plan for the Development of National Animal Husbandry (2011–2015)* and the *National Pig Production Development Plan (2016–2020)* issued by the Ministry of Agriculture of China in 2006, 2011 and 2016, respectively, are long-term plans, which provide for the development of each five-year. The plans lay out the development path of the industry and

provide goals and directions for the development vision of the hog breeding industry, which have an important guiding role for the industry. Examining the GPE of the hog breeding industry in different planning periods can reflect the development characteristics of the industry led by policies, so this paper divides the sample period into I-period (2006–2010), II-period (2011–2015) and III-period (2016–2019).

(due to some missing data, the 2020 GPE was not calculated). Data were primarily obtained from the *Compilation of Information on the Cost and Benefit of Agricultural Products in China*, *China Agricultural Statistical Yearbook*, *China Rural Statistical Yearbook*, *China Environmental Statistical Yearbook*, and EPS database from 2005 to 2020. Price-related data were adjusted for inflation using the price index of agricultural production materials (with 2006 as the base period) to ensure accurate comparisons.

Moreover, considering various factors such as product development base, environmental impact, resource allocation, consumer preference, slaughter, and processing, the Ministry of Agriculture has released *the China Pig Production Development Plan (2016–2020)*, which divides the country into four pig development regions: key development region, constrained development region, potential growth region, and moderate development region*China Pig Production Development Plan* (2016–2020) issued by the Ministry of Agriculture and Rural Affairs of the People's Republic of China divides the pig development regions, with the key development region including Hebei, Shandong, Henan, Chongqing, Guangxi, Sichuan, Hainan; the constrained development region including Beijing, Tianjin, Shanghai, Jiangsu, Zhejiang, Fujian, Anhui, Jiangxi, Hubei, Hunan and Guangdong; the potential growth region including Liaoning, Jilin, Heilongjiang, Inner Mongolia, Yunnan and Guizhou; and the moderate development region including Shanxi, Shaanxi, Gansu, Xinjiang, Tibet, Qinghai, and Ningxia.

. In this paper, we further extend this classification to analyze the spatial divergence and evolution of GPE.

The data utilized in this study are official statistics and do not involve human participants or raise ethical issues. Therefore, this study was not subject to review and approval by an institutional review board (ethics committee), and participant consent was not obtained.

## Variable settings

According to the arrangement of the study, this section selects the input-output variables required to measure GPE and the driving factors variables that may affect the spatial divergence of GPE to provide material support for the study. presented in Table 1.

**Input-output variables.** When selecting variables, it is recommended to prioritize physical quantities whenever possible [12]. In addition, to ensure scientific and comparable results, it is important to choose variables that are consistently measured across regions and less influenced by market price fluctuations [26]. Regarding input indicators, previous research has typically categorized them into two types: conventional inputs and resource inputs. The key costs of traditional hog breeding include labor, piglets, and feed. Therefore, this study uses labor input, piglet input, and feed input as conventional input variables to measure the production efficiency of the hog breeding industry. To comprehensively evaluate the green production capacity of hog breeding, this study also incorporates water and energy—both closely linked to sustainable development—as input variables. Specifically, water input and energy input are included in the analysis.

The selected desirable output indicator is the net main product yield (net hog weight gain), which is the difference between the main product yield of hogs and the weight of piglets. In regards to the selection of undesirable output indicators, both carbon emissions and non-point

**Table 1. Input-output and driving factors variables.**

| Indicators | Variables | Description of Variables | Reference |
|---|---|---|---|
| Conventional input | Labor input | Labor cost per unit (USD) | Zhao et al. [12] |
| | Piglet input | Piglet cost per unit (USD) | Zhao et al. [12] |
| | Feed input | Concentrated feed cost per unit (USD) | Zhao et al. [12] |
| Resource input | Energy input | Fuel power cost per unit (USD) | Zhong et al. [11] |
| | Water input | Feeding water cost per unit (USD) | Zhong et al. [11] |
| Desirable output | Net hog weight gain | Weight of hog product-weight of piglets (kg) | Zhao et al. [12] |
| Undesirable output | Non-point source pollutants Emissions | Non-point source pollutants (10 kt) | Streimikis & Saraji [31] |
| | Carbon emissions | Carbon emissions (10 kt) | Zhu et al. [24] |
| Driving Factors | Scale breeding | Number of large-scale hog farms/total number of farms (%) | Zhong et al. [11] |
| | Environmental load bearing | Hog stock/ cultivated land (head/m$^2$) | Zheng et al. [25] |
| | Production operation capability | Number of slaughtered fattened hogs/quantity of live hogs at the beginning of the year (%) | Wang et al. [47] |
| | Resource endowment | Maize yield (10 kt) | Wang et al. [47] |
| | Environmental investment | Environmental governance investment (100 million USD) | Wang et al. [47] |

source pollutants are deemed as undesirable outputs of hog breeding [27]. While some scholars choose carbon emission [28], others prefer the amount of non-point source pollutants [29]. This paper measures both carbon emissions and non-point source pollutants by drawing on Zhou et al. [30] and Streimikis & Saraji [31]. To accurately estimate GPE, this paper considered as many pollutants as possible, and finally selected COD, total nitrogen (TN), and total phosphorus (TP) as non-point source pollutants. The pollutants emitted from the hog breeding process were then converted into carbon equivalents to measure their total carbon emissions.

According to the study by Zhang et al. [32], the emissions of non-point source pollutants can be calculated by the following formulas:

$$COD = N \times T \times (G_m \times G_{m-COD} + G_u \times G_{u-COD}) \tag{1}$$

$$TN = N \times T \times (G_m \times G_{m-TN} + G_u \times G_{u-TN}) \tag{2}$$

$$TP = N \times T \times (G_m \times G_{m-TP} + G_u \times G_{u-TP}) \tag{3}$$

where, $COD$, $TN$, $TP$ represent COD, TN and TP emission, $N$ stands for hog breeding quantity, measured by annual hog output, $T$ represents feeding cycle, $G_m$ and $G_u$ represent daily emission coefficient of hog manure and hog urine respectively, $G_{m-COD}$ and $G_{u-COD}$ represent COD emission coefficient of hog manure and hog urine respectively, $G_{m-TN}$ and $G_{u-TN}$ represent TN emission coefficient of hog manure and hog urine, $G_{m-TP}$ and $G_{u-TP}$ represent TP emission coefficient of hog manure and hog urine.

Referring to Yao et al. [33], the carbon emissions from hog breeding are measured based on the whole life cycle with the following equations:

$$TC_{sc} = APP \times (\frac{cost_e}{price_e} \times ef_e + \frac{cost_c}{price_c} \times ef_c) \tag{4}$$

$$TC_{sg} = P \times \frac{MJ}{e} \times ef_e \tag{5}$$

$$APP = T \times \frac{N}{365} \tag{6}$$

where, $TC_{sc}$ and $TC_{sg}$ represent the carbon emission of feeding process and processing process respectively, $APP$ is the average annual feeding quantity of hogs, $cost_e$ and $cost_c$ represent the electricity expenditure and coal expenditure per unit of hog breeding respectively, $price_e$ and $price_c$ represent the unit price of electricity and coal respectively, $ef_e$ and $ef_c$ are the $CO_2$ emission coefficient of electricity consumption and coal consumption. $P$ represents the annual output of pork, $MJ$ is the energy dissipation coefficient of pork per unit processing, and $e$ is the heat value generated by consuming one unit of electricity.

$$TC_{CH_4} = APP \times (ef_{i-CH_4} + ef_{j-CH_4}) \times GWP_{CH_4} \tag{7}$$

$$TC_{N_2O} = APP \times ef_{j-N_2O} \times GWP_{N_2O} \tag{8}$$

$$TC = (TC_{sc} + TC_{sg} + TC_{CH_4} + TC_{N_2O}) \times e_{tpf} \tag{9}$$

where, $TC_{CH_4}$, $TC_{N_2O}$ and $TC$ represent $CH_4$ global warming potential, $N_2O$ global warming potential and total carbon emission of hog breeding industry respectively, $ef_{i-CH_4}$, $ef_{j-CH_4}$ and $ef_{j-N_2O}$ represent $CH_4$ emission coefficient of pig gastrointestinal fermentation, $CH_4$ and $N_2O$ emission coefficient of manure management respectively. $GWP_{CH_4}$ and $GWP_{N_2O}$ are $CH_4$ and $N_2O$ global warming potential respectively, and $e_{tpf}$ is the conversion of $CO_2$ equivalent is converted to standard carbon coefficient. Table 2 shows the meaning of various emission coefficients and their values.

**Driving factors variables.** This paper has selected several independent variables, including scale breeding, environmental load bearing, production and operation capacity, resource endowment, and environmental investment, to investigate their impact on the spatial divergence of GPE in hog breeding. The scale of hog breeding is closely linked to its standardization capacity, which develops in tandem [34]. Standardization is a vital prerequisite for achieving green and high-quality development in the hog breeding industry. Hence, scale breeding is a crucial factor affecting GPE in this industry, and differences in scale breeding across regions will exacerbate the spatial divergence in GPE.

The environmental load bearing refers to the number of hogs per unit of arable land, and mitigating the environmental carrying capacity is a crucial measure for improving the industry's green efficiency and sustainable development [35]. The higher the regional environmental loading intensity, the greater the number of hogs bred, the more significant the impact on breeding activities and the level of environmental management. Therefore, environmental load bearing is an important variable included in the system of driving factors.

The hog output rate, a pivotal indicator of hog production and capacity [36], is a significant reflection of the industry's production and operational capacity. Developing production and operational capacity constitutes an effective approach to enhance the industry's green

**Table 2. Hog breeding pollutant emission coefficient.**

| | Emission coefficient | Symbol | Value | Units |
|---|---|---|---|---|
| **Non-point source pollutants emission coefficients** | Feeding cycle | T | 199 | d |
| | Daily emission coefficient of hog manure | $G_m$ | 2.00 | kg/d |
| | Daily emission coefficient of hog urine | $G_u$ | 3.30 | kg/d |
| | COD emission coefficient of hog manure | $G_{m\text{-}COD}$ | 52.00 | kg/t |
| | TN emission coefficient of hog manure | $G_{m\text{-}TN}$ | 5.88 | kg/t |
| | TP emission coefficient of hog manure | $G_{m\text{-}TP}$ | 3.41 | kg/t |
| | COD emission coefficient of hog urine | $G_{u\text{-}COD}$ | 9.00 | kg/t |
| | TN emission coefficient of hog urine | $G_{u\text{-}TN}$ | 3.30 | kg/t |
| | TP emission coefficient of hog urine | $G_{u\text{-}TP}$ | 0.52 | kg/t |
| **Carbon emission coefficients** | Unit price of electricity | $Price_e$ | 0.062 | USD/kW·h |
| | Unit price of coal | $Price_c$ | 116 | USD/t |
| | $CO_2$ emission coefficient of electric consumption | $ef_e$ | 0.973 | tCO2/MW·h |
| | $CO_2$ emission coefficient of coal consumption | $ef_c$ | 1.98 | t/t |
| | Energy dissipation coefficient per unit processing of pork | MJ | 3.76 | MJ/kg |
| | Heat value generated by consuming one unit of electricity | e | 3.60 | MJ |
| | $CH_4$ emission coefficient of hog gastrointestinal fermentation | $ef_{i\text{-}CH4}$ | 1.00 | kg/ head |
| | $CH_4$ emission coefficient of hog manure management | $ef_{j\text{-}CH4}$ | 3.50 | kg/ head |
| | $N_2O$ emission coefficient of hog manure management | $ef_{j\text{-}N2O}$ | 0.530 | kg/ head |
| | $CH_4$ global warming potential | $GWP_{CH4}$ | 21 | |
| | $N_2O$ global warming potential | $GWP_{N2O}$ | 310 | |
| | $CO_2$ equivalent is converted to standard carbon coefficient | $e_{tpf}$ | 0.2728 | |

Note: The emission coefficient of non-point source pollutants is taken from the statistics released by the Ministry of Ecology and Environment of China (https://www.mee.gov.cn). Carbon emission coefficient is determined in conjunction with IPCC guidelinesIPCC. IPCC Guidelines for National Greenhouse Gas Inventories Volume 4: Agriculture, Forestry and Other Land Use[R]. Geneva: IPCC, 2006

development capacity [37]. Consequently, the variation in production and operation capacity emerges as a crucial factor contributing to the spatial divergence in GPE of the hog breeding industry.

Resource endowment exerts a direct and positive influence on the industry's green development efficiency [38] and constitutes the fundamental basis for enhancing the GPE of the hog breeding industry. The differences in resource endowments among regions may lead to divergences in GPE. Additionally, since maize is the primary raw material for hog feed, a higher maize yield corresponds to stronger green farming capacity for hogs. Thus, this paper employs Kolleen & Norman's [39] methodology to gauge feed production by maize yield, which in turn represents the resource endowment status of the hog farming industry in each province, and scrutinizes its impact on the spatial divergence of GPE.

The effectiveness of environmental management practices is heavily influenced by the level of environmental investment made in the region, with higher investments indicating a greater emphasis on environmental management [40]. This highlights the importance of promoting green hog breeding practices, which can significantly enhance the industry's GPE. As a result, differences in the environmental management status of each region can impact the spatial divergence of the hog breeding industry's GPE across regions.

## Methods

This section provides a detailed description of the methods to be applied in this study, including the SBM model, the divergence index, and the Geodetector, which helps to improve the applicability of the study.

**SBM model.** To measure the GPE of the hog breeding industry in each province, the SBM model based on undesirable outputs is utilized. The non-radial and non-angular SBM directional distance function, proposed by Tone [41], fully addresses the input-output slackness problem. This method directly incorporates slack variables into the objective function to solve the non-zero input-output slackness issue, which eliminates the non-efficiency factors caused by slackness. Furthermore, the SBM model is dimensionless and non-angular, thus avoiding bias and effects of different magnitudes and angle selection differences. The SBM model considers $m$ inputs (x), $n_1$ desirable outputs ($y^g$), and $n_2$ undesirable outputs ($y^b$) and can be expressed as:

$$Min\rho = \frac{1 - \frac{1}{m}\sum_{i=1}^{m}\frac{s_i^-}{x_{i0}}}{1 + \frac{1}{n_1+n_2}\left(\sum_{r=1}^{n1}\frac{s_r^g}{y_{r0}^g} + \sum_{r=1}^{n2}\frac{s_r^b}{y_{r0}^b}\right)} \tag{10}$$

$$s.t. \begin{cases} x_0 = X\lambda + s^- \\ y_0^g = Y^g\lambda - s^g \\ y_0^b = Y^b\lambda + s^b \\ \lambda \geq 0, s^- \geq 0, s^g \geq 0, s^b \geq 0, \end{cases} \tag{11}$$

where $s$ represents the slack in inputs and outputs, $s^-$ indicates too many inputs, $s^b$ indicates too many undesirable outputs, $s^g$ indicates insufficient desirable outputs, $\lambda$ indicates the weights, $\rho(0 \leq \rho \leq 1)$ indicates the attainment efficiency score, and the rates of $s^-$, $s^b$ and $s^g$ are strictly decreasing. When $s^-$, $s^b$ and $s^g$ are all equal to 0, that is, when $\rho=1$, it means that there is no excess of inputs and undesirable outputs, and there is no deficit of desirable outputs. Thus, the DMU is completely efficient. However, when $s^-$, $s^b$ and $s^g$ are all greater than 0, that is, when $\rho<1$, it means that there is an efficiency loss in DMU. The output level can be maintained by reducing the input and undesirable output, indicating that the DMU is invalid.

**Divergence index.** The measurement of regional divergence in development levels is commonly done using methods such as the Gini coefficient, the Theil index, and the mean log deviation (generalized entropy index). These methods are sensitive to changes in high, medium, and low levels, respectively, and their results are complementary. Thus, scholars usually analyze regional divergence in development levels based on the comparison results of these three methods [42, 43].

The modified Gini coefficient is widely used in industrial economics and is one of the main methods for measuring differences in the level of industrial development. There are various methods for measuring the Gini coefficient, and this paper adopts the method proposed by Mookherjee and Shorrocks [44]. The basic formula for this method is:

$$GINI = \frac{1}{2n^2\mu}\sum|GPE_i - GPE_j| \tag{12}$$

where $n$ denotes the number of provinces, $GPE_i$ and $GPE_j$ represent the GPE of the hog

breeding industry in province $i$ and province $j$, respectively, and $\mu$ is the average value of GPE in each province.

The basic formulae for the mean log deviation ($GE_0$) and the Thiel index ($GE_1$) are as follows:

$$GE_0(GPE) = \frac{1}{n}\sum_{i\in N}ln\frac{\mu}{GPE_i} \tag{13}$$

$$GE_1(GPE) = \frac{1}{n}\sum_{i\in N}\frac{GPE_i}{\mu}ln\frac{GPE_i}{\mu} \tag{14}$$

where $n$ denotes the number of provinces, $\mu$ is the average value of GPE of the hog breeding industry in each province and $GPE_i$ denotes the level of GPE in province $i$.

To explore the trend of inter-provincial differences over time, this paper further examines the convergence of GPE. $\alpha$ convergence is used to analyze the discrete trends in GPE of the national and regional hog breeding industry. The $\alpha$ convergence formula is:

$$\alpha_t = \sqrt{n^{-1}\sum_{i=1}^{n}\left\{GPE_i(t) - \left[n^{-1}\sum_{m=1}^{n}GPE_m(t)\right]\right\}^2} \tag{15}$$

where $GPE_i(t)$ denotes the GPE of the hog breeding industry in period $t$ of the $i$ province, $GPE_m(t)$ denotes the GPE in period $t$ of the $m$ province, and $n$ denotes the number of provinces. The decreasing value of $\alpha_t$ indicates that the GPE in the sample period is converging and the difference between provinces is narrowing, while the increasing of $\alpha_t$ indicates divergence, and the difference in GPE among provinces is enlarged.

**Geodetector.** The Geographic comprises of four key components, namely the factor detector, interaction detector, risk detector, and ecological detector. The fundamental assumption of this approach is that the spatial distribution of the driving factors responsible for the change is in agreement with the spatial distribution of that particular phenomenon. It suggests that the driving factors significantly impact the spatial divergence of the phenomenon under study [45]. This study investigates the driving factors of spatial divergence of GPE through factor detector and interaction detector. The factor detector is utilized to examine the impact of a factor on the spatial divergence of a variable, whereas the interaction detector is employed to explore the effect of factor interactions on the variable. By developing a novel spatial layer comprising two driving factors, the interaction detector assesses the impact of two factors and their superimposed layers on spatial divergence. The primary aim of the interaction detector is to unveil the interaction between different factors on spatial divergence and to compare the interaction effect with that of a single factor on the variables. The factor detector assesses whether factors affect the spatial divergence of variables by comparing whether the spatial distribution of driving factors and variables is consistent. The degree of factor detector is measured using the q-statistic, expressed as:

$$q = 1 - \frac{\sum_{h=1}^{H}N_h\sigma_h^2}{N\sigma^2} \tag{16}$$

$q$-statistic denotes the role of factors driving the spatial divergence of GPE, with q ranging from 0 to 1. $N$ denotes the regional sample size. $H$ is the partition of factors and variables and denotes the sample size of sub-region $h$. $q = 0$ means that the spatially stratified heterogeneity is not influenced by the factor, and $q = 1$ means that the spatial divergence is completely influenced by the factor. The larger the $q$-statistic, the greater the influence of the factor on the

spatial divergence of GPE. In this regard, the *q*-statistic indicates that the spatial divergence of factors explains the spatial divergence of variables, and it does not require a significance test [46]. To process data, all driving factors must be transformed into quantitative variables. This study uses the Q-type clustering analysis method to cluster independent variables into six categories according to numerical size, and observations are grouped into six types from high to low. Finally, the Geodetector is used to calculate the degree of influence of each factor on the spatial divergence of GPE in the hog breeding industry.

## Results

This chapter employs the aforementioned materials and methods to gauge the GPE of the hog breeding industry and conducts an analysis of the outcomes. It expounds upon the development characteristics and evolutionary trends of GPE in diverse regions and provinces across various time periods, examining both spatial and temporal dimensions, while offering profound interpretations for the underlying reasons. The findings of this chapter serve as the foundation for scrutinizing the spatial divergence of GPE and its driving factors.

### Statistical analysis of variables

Table 3 presents the results of the descriptive statistical analysis of input and output variables. The analysis revealed that, firstly, conventional input variables witnessed a significant increase as compared to 2006, while resource input variables remained stable, and the net weight gain of hogs also increased significantly. This suggests that hog production capacity increased with the increase in resource input. Secondly, there was a significant decrease in non-point source pollutant emissions, whereas carbon emissions increased initially and then decreased, indicating the considerable effect of pollutant emission reduction during the hog breeding process.

### Measurement of the GPE

The SBM model was used to calculate the mean values of GPE in the hog breeding industry in the key development region, constrained development region, potential growth region, and moderate development region in the I, II, and III periods. The results are shown in Fig 1.

**Table 3. Descriptive statistics of input-output variables.**

| Year | Labor input (USD) | Piglet input (USD) | Feed input (USD) | Energy input (USD) | Water input (USD) | Net hog weight gain(kg) | COD (billion tons) | TD (billion tons) | TP (billion tons) | Carbon emission (billion tons) |
|---|---|---|---|---|---|---|---|---|---|---|
| 2006 | 4.559 | 24.258 | 54.878 | 0.892 | 0.233 | 81.3 | 0.134 | 0.023 | 0.0086 | 10.51 |
| 2007 | 5.894 | 43.353 | 63.409 | 0.906 | 0.281 | 84.2 | 0.132 | 0.022 | 0.0084 | 9.65 |
| 2008 | 6.106 | 58.074 | 73.636 | 0.955 | 0.295 | 86.7 | 0.117 | 0.020 | 0.0075 | 11.73 |
| 2009 | 8.412 | 51.501 | 95.254 | 1.162 | 0.344 | 88.11 | 0.123 | 0.021 | 0.0079 | 12.43 |
| 2010 | 9.253 | 46.309 | 100.219 | 1.080 | 0.344 | 89.64 | 0.125 | 0.021 | 0.0080 | 12.19 |
| 2011 | 11.344 | 70.028 | 108.882 | 1.076 | 0.341 | 91.51 | 0.124 | 0.021 | 0.0079 | 12.79 |
| 2012 | 13.304 | 78.966 | 128.491 | 1.152 | 0.369 | 93.65 | 0.124 | 0.021 | 0.0079 | 13.68 |
| 2013 | 15.764 | 78.554 | 142.222 | 1.146 | 0.394 | 93.86 | 0.127 | 0.021 | 0.0081 | 13.66 |
| 2014 | 16.442 | 73.195 | 149.258 | 1.291 | 0.431 | 95.81 | 0.126 | 0.021 | 0.0081 | 14.98 |
| 2015 | 16.412 | 77.212 | 138.169 | 1.217 | 0.417 | 96.31 | 0.124 | 0.021 | 0.0079 | 14.97 |
| 2016 | 16.445 | 107.976 | 125.459 | 1.153 | 0.394 | 98.86 | 0.120 | 0.020 | 0.0077 | 14.69 |
| 2017 | 16.079 | 95.468 | 120.178 | 1.194 | 0.401 | 99.91 | 0.116 | 0.020 | 0.0074 | 16.34 |
| 2018 | 16.218 | 70.318 | 123.615 | 1.230 | 0.428 | 101.56 | 0.117 | 0.020 | 0.0075 | 16.53 |
| 2019 | 15.960 | 92.559 | 122.414 | 1.135 | 0.399 | 105.47 | 0.114 | 0.019 | 0.0073 | 13.46 |

As shown in Fig 1, the overall GPE increased to an average of 0.401, indicating an improvement in the industry's efficiency. The highest GPE was observed in the I-period, followed by a decrease in the II-period, and a recovery in the III-period. Regionally, the key development region had the highest overall GPE of 0.412, followed by the constrained and moderate development regions, and the potential growth region had the lowest overall GPE of 0.374. In addition, except for the constrained development region, the GPE of the remaining regions grew during the sample period, with the fastest growth rate in the potential growth region, with an average GPE of 0.411 in the III-period, an increase of 10.5% compared to the I-period. Followed by the key development region, with an average 5.8% increase in GPE, and the moderate development region was stable overall, while the constrained development region decreased by 10.8% in GPE. Nine provinces, including Beijing, Tianjin, Shanxi, Inner Mongolia, Liaoning, Jilin, Shanghai, Guangxi, and Gansu, experienced a decline in GPE throughout the sample period, while the remaining 20 provinces showed an increase. Most provinces exhibited an increase in GPE in the I-period and III-period, and a decrease in the II-period.

Fig 2 depicts the chronological progression of the GPE in China's hog breeding industry. Nationally, the GPE observed a substantial increase from 0.409 to 0.496. GPE displayed an initial incline and subsequent decline during the I-period, reaching its highest value in 2008. The

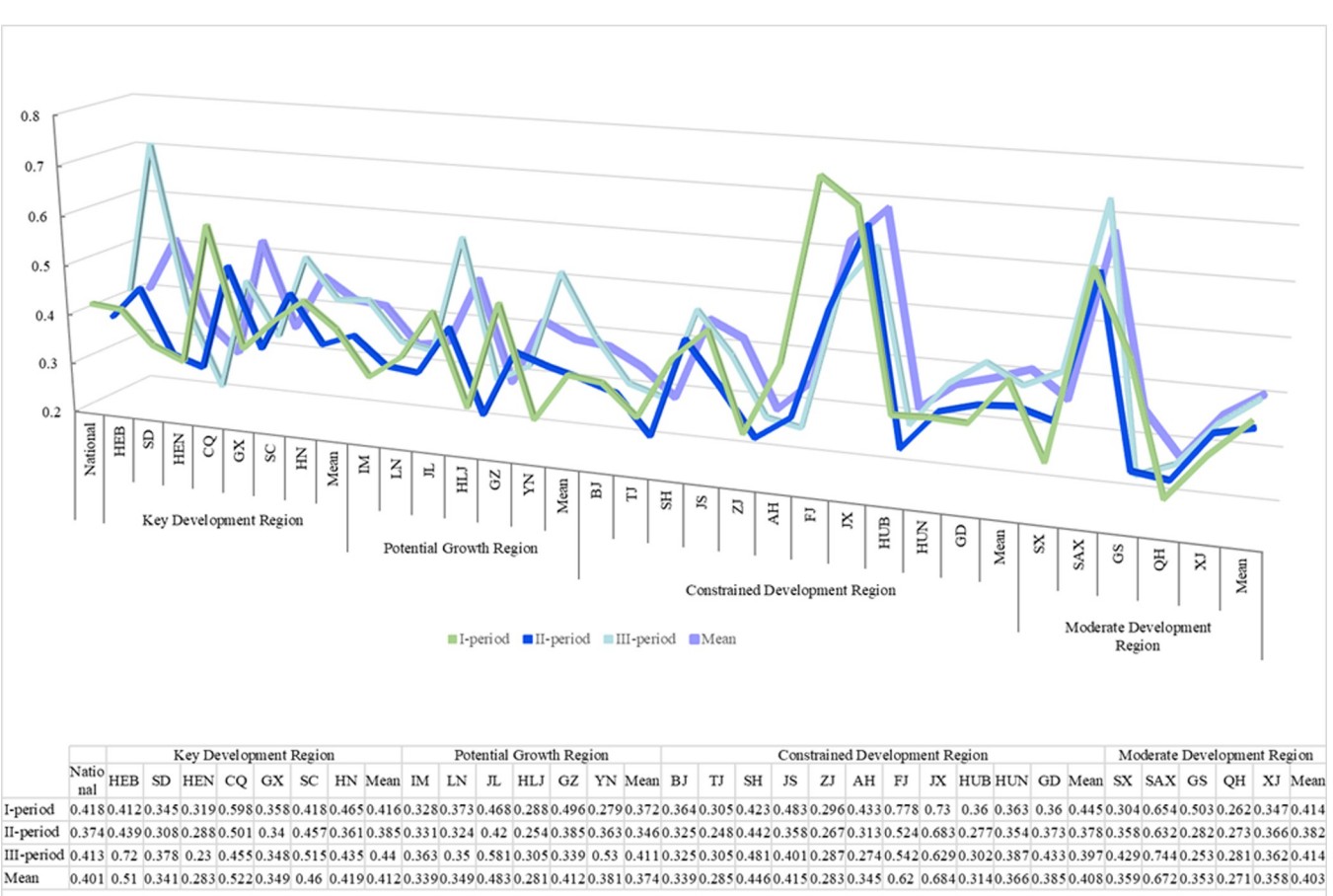

| | National | Key Development Region | | | | | | | | Potential Growth Region | | | | | | | Constrained Development Region | | | | | | | | | | | | Moderate Development Region | | | | | |
|---|---|---|---|---|---|---|---|---|---|---|---|---|---|---|---|---|---|---|---|---|---|---|---|---|---|---|---|---|---|---|---|---|---|---|
| | | HEB | SD | HEN | CQ | GX | SC | HN | Mean | IM | LN | JL | HLJ | GZ | YN | Mean | BJ | TJ | SH | JS | ZJ | AH | FJ | JX | HUB | HUN | GD | Mean | SX | SAX | GS | QH | XJ | Mean |
| I-period | 0.418 | 0.412 | 0.345 | 0.319 | 0.598 | 0.358 | 0.418 | 0.465 | 0.416 | 0.328 | 0.373 | 0.468 | 0.288 | 0.496 | 0.279 | 0.372 | 0.364 | 0.305 | 0.423 | 0.483 | 0.296 | 0.433 | 0.778 | 0.73 | 0.36 | 0.363 | 0.36 | 0.445 | 0.304 | 0.654 | 0.503 | 0.262 | 0.347 | 0.414 |
| II-period | 0.374 | 0.439 | 0.308 | 0.288 | 0.501 | 0.34 | 0.457 | 0.361 | 0.385 | 0.331 | 0.324 | 0.42 | 0.254 | 0.385 | 0.363 | 0.346 | 0.325 | 0.248 | 0.442 | 0.358 | 0.267 | 0.313 | 0.524 | 0.683 | 0.277 | 0.354 | 0.373 | 0.378 | 0.358 | 0.632 | 0.282 | 0.273 | 0.366 | 0.382 |
| III-period | 0.413 | 0.72 | 0.378 | 0.23 | 0.455 | 0.348 | 0.515 | 0.435 | 0.44 | 0.363 | 0.35 | 0.581 | 0.305 | 0.339 | 0.53 | 0.411 | 0.325 | 0.305 | 0.481 | 0.401 | 0.287 | 0.274 | 0.542 | 0.629 | 0.302 | 0.387 | 0.433 | 0.397 | 0.429 | 0.744 | 0.253 | 0.281 | 0.362 | 0.414 |
| Mean | 0.401 | 0.51 | 0.341 | 0.283 | 0.522 | 0.349 | 0.46 | 0.419 | 0.412 | 0.339 | 0.349 | 0.483 | 0.281 | 0.412 | 0.381 | 0.374 | 0.339 | 0.285 | 0.446 | 0.415 | 0.283 | 0.345 | 0.62 | 0.684 | 0.314 | 0.366 | 0.385 | 0.408 | 0.359 | 0.672 | 0.353 | 0.271 | 0.358 | 0.403 |

**Fig 1. GPE of the hog breeding industry.** Note: HEB, SD, HEN, CQ, GX, SC, HN, IM, LN, JL, HLJ, GZ, YN, BJ, TJ, SH, JS, ZJ, AH, FJ, JX, HUB, HUN, GD, SX, SAX, GS, QH, XJ respectively stand for Hebei province, Shandong province, Henan province, Chongqing, Guangxi province, Sichuan province, Hainan province, Inner Mongolia province, Liaoning province, Jilin province, Heilongjiang province, Guizhou province, Yunnan province, Beijing, Shanghai, Jiangsu province, Zhejiang province, Anhui province, Fujian province, Jiangxi province, Hubei province, Hunan province, Guangdong province, Shanxi province, Shaanxi province, Gansu province, Qinghai province, and Xinjiang province.

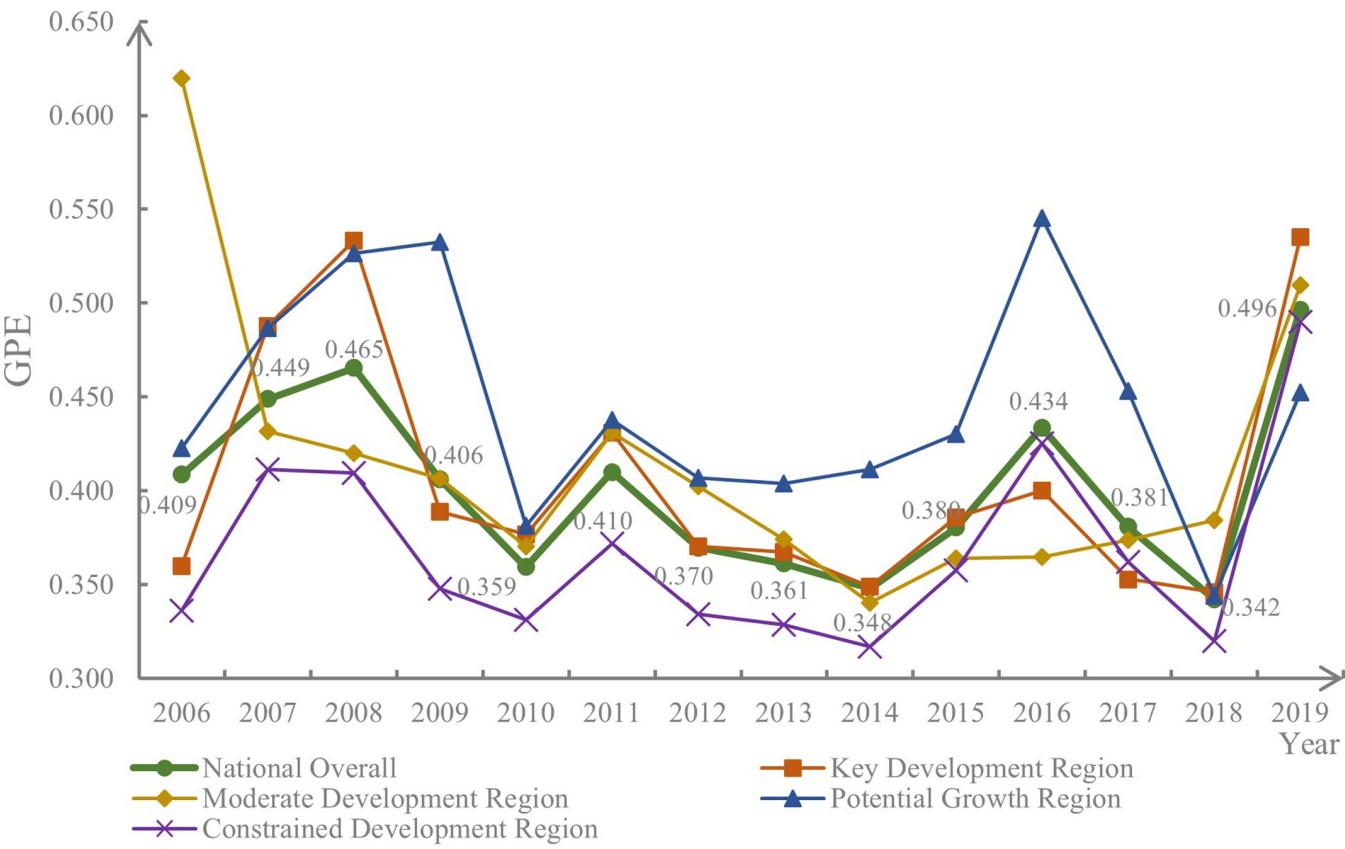

**Fig 2. Temporal evolution of GPE in the hog breeding industry.**

cause was mainly attributed to the outbreak of Porcine Reproductive and Respiratory Syndrome (PRRS) in hogs from 2007 to 2008, leading to a persistent decline in hog population and a consequent decrease in hog production capacity. During the II-period, GPE experienced a slight drop between 2011 and 2013, followed by a gradual improvement in 2014. The main reason for the decline was the lean meat powder scandal in 2011, which affected the hog production efficiency. However, the *Regulations on Prevention and Control of Pollution in Livestock and Poultry Scale Farming* promulgated in 2014 stimulated the green development of the hog breeding industry, thereby leading to the recovery of GPE. During the III-period, GPE witnessed an initial decline and subsequent increase, attaining its lowest point of 0.342 in 2018, followed by a sharp rise in 2019. This was primarily due to the impact of African Swine Fever (ASF) in 2018, which significantly reduced hog breeding efficiency and spurred the rapid reorganization of the industry, culminating in the enhanced green development of the hog breeding industry and a corresponding surge in hog GPE in 2019.

At the regional level, the GPE in the potential growth region exhibited cyclical fluctuations with a period of five years, and had the highest value overall. The GPE first increased and then decreased in the I-period, remained stable during the II-period, and increased again in the III-period. Due to the abundance of resources in the region, the production efficiency of hog breeding was high, and showed an upward trend. In the moderate development region, the GPE experienced a rapid decline from 2006 to 2007, followed by a sharp increase from 2018 to 2019, with a relatively stable period in between. This region experienced less breeding pressure and had a high level of green development overall, with little impact from the hog breeding

industry on its economic development and market stability. In the key development region, the GPE of the hog breeding industry first increased and then decreased in the I-period, remained relatively stable during the II-period, and experienced a decrease followed by an increase in the III-period. This region had a large proportion of the hog breeding industry, and was greatly influenced by animal diseases and government regulations, resulting in a fluctuating trend. The overall GPE of hog breeding in the constrained development region was the lowest, with a fluctuating trend in development. This region was economically developed, with a dense water network and limited hog breeding, resulting in little influence from various factors on the GPE.

## Spatial divergence

Building upon the aforementioned measurement results, this section proceeds to delve into an in-depth analysis of the extent of divergence in GPE within various regions and provinces throughout the sample period. Expanding on this foundation, the dispersion trend of GPE divergence among different regions is explored through the implementation of an α convergence test, while elucidating the underlying reasons behind this evolving trend. Moreover, this chapter examines the distinctive features of spatial divergence in GPE within the hog breeding sector across different periods and regions, affirming the driving factors through the scrutiny of input-output factors and external factors.

In this paper, Gini coefficient, Theil index and mean log deviation were used to explore the divergence of hog breeding industry GPE in different regions during the sample period. The results are shown in Table 4.

The Gini coefficient, Theil index, and mean log deviation all displayed a consistent pattern, revealing that the spatial divergence of GPE initially decreased and then widened, ultimately leading to an overall expansion. The I-period demonstrated a conspicuous trend of narrowing spatial divergence, with the values of the three indicators dropping from 0.107, 0.078, and 0.089 in 2006 to 0.061, 0.023, and 0.023 in 2010. The spatial divergence experienced a slight increase and remained relatively stable in the II-period. In the III-period, the spatial divergence initially decreased and then increased, reaching its lowest point in 2018 with values of 0.076, 0.037, and 0.038 for the three indicators, and then rapidly expanding in 2019. The international community generally regards a Gini coefficient of 0.3 or less as indicating a small gap in the level of inequality. The results demonstrate that, although China's hog breeding industry's GPE did not significantly vary from 2006 to 2019, the inter-provincial spatial divergence exhibited a fluctuating trend.

Table 4 examines the spatial divergence of GPE across regions during the sample period, using the Gini coefficient as a measure. Looking at the mean values, the potential growth region had the highest spatial divergence, with mean values of 0.104 and 0.089, followed by the moderate development region and key development region, while the constrained development region had the lowest spatial divergence. In terms of changes in different regions, the overall spatial divergence of GPE remained stable in the key development region and continued to increase in the constrained development region. The spatial divergence in the moderate development region remained stable during the I-period and II-period but increased rapidly during the III-period. In the potential growth region, the spatial divergence decreased initially and then increased, with the smallest spatial divergence in the II-period. It can be observed that the divergence in the green production capacity of hogs between provinces within each region has expanded.

Fig 3 depicts the discrete trends of the spatial divergence of GPE nationwide, using the Gini coefficient as an example, with the α convergence test. The spatial divergence of GPE in China

**Table 4. Divergence of GPE in the hog breeding industry.**

| Year | GINI | $GE_0$ | $GE_1$ | Key Development Region GINI | Moderate Development Region GINI | Potential Growth Region GINI | Constrained Development Region GINI |
|------|------|-----|-----|------|------|------|------|
| 2006 | 0.107 | 0.078 | 0.089 | 0.034 | 0.145 | 0.114 | 0.024 |
| 2007 | 0.089 | 0.051 | 0.054 | 0.054 | 0.080 | 0.143 | 0.055 |
| 2008 | 0.101 | 0.065 | 0.070 | 0.105 | 0.077 | 0.138 | 0.053 |
| 2009 | 0.091 | 0.058 | 0.065 | 0.077 | 0.078 | 0.133 | 0.033 |
| 2010 | 0.061 | 0.023 | 0.023 | 0.062 | 0.060 | 0.062 | 0.042 |
| I-period | 0.090 | 0.055 | 0.060 | 0.066 | 0.088 | 0.118 | 0.041 |
| 2011 | 0.074 | 0.035 | 0.036 | 0.062 | 0.073 | 0.090 | 0.052 |
| 2012 | 0.073 | 0.035 | 0.036 | 0.057 | 0.086 | 0.088 | 0.053 |
| 2013 | 0.070 | 0.032 | 0.034 | 0.059 | 0.082 | 0.089 | 0.043 |
| 2014 | 0.081 | 0.042 | 0.045 | 0.058 | 0.088 | 0.111 | 0.051 |
| 2015 | 0.079 | 0.040 | 0.040 | 0.051 | 0.107 | 0.086 | 0.061 |
| II-period | 0.075 | 0.037 | 0.038 | 0.057 | 0.087 | 0.093 | 0.052 |
| 2016 | 0.102 | 0.069 | 0.071 | 0.055 | 0.126 | 0.116 | 0.083 |
| 2017 | 0.087 | 0.049 | 0.051 | 0.074 | 0.086 | 0.092 | 0.076 |
| 2018 | 0.076 | 0.037 | 0.038 | 0.062 | 0.094 | 0.079 | 0.060 |
| 2019 | 0.108 | 0.075 | 0.076 | 0.071 | 0.131 | 0.110 | 0.107 |
| III-period | 0.093 | 0.058 | 0.059 | 0.066 | 0.109 | 0.099 | 0.082 |
| Average 1 | 0.086 | 0.049 | 0.052 | 0.063 | 0.094 | 0.104 | 0.057 |
| Average 2 | 0.072 | 0.033 | 0.035 | 0.055 | 0.083 | 0.089 | 0.051 |

Note: average 1 is the mean value of the three indicators from 2006 to 2019, and average 2 is the value of the three indicators measured based on the mean value of GPE in each province from 2006 to 2019.

displays α discrete evolution. Except for the moderate development region, which displays convergence evolution as a whole, the α-convergence values of all other regions expand, indicating α discrete evolution. The potential growth region registers the largest value of α but the slowest convergence rate, whereas the constrained development region has the smallest value of α but the fastest convergence rate. The convergence trend varies across periods. In the I-period, the spatial divergence of GPE in the moderate development and potential growth regions displays α discrete, while the key development and constrained development regions show α convergence. During the II-period, the regions remain stable at a lower level of convergence, without significant fluctuations of spatial divergence. In the III-period, the moderate development, potential growth, and constrained development regions evolve in convergence, followed by discretization, whereas the key development region continues in a discrete trend.

Furthermore, the significant fluctuations in convergence during the years 2010, 2016, and 2018 were linked to the adjustments made in industrial development efforts across each region. Between 2008 and 2009, the Chinese government strongly advocated for rural environmental protection and promoted large-scale hog breeding, which led to an increase in the green production capacity of hogs in each province and a reduction in the gap between them in 2010. In 2014, the implementation of the *Regulation on the Prevention and Control of Pollution from Large-scale Breeding of Livestock and Poultry* prompted provincial governments to adjust the direction of industrial development. However, due to differences in the provincial baseline and response measures, spatial divergence in GPE among provinces in the hog breeding industry continued to expand. In 2018, the outbreak of African Swine Fever (ASF) had a

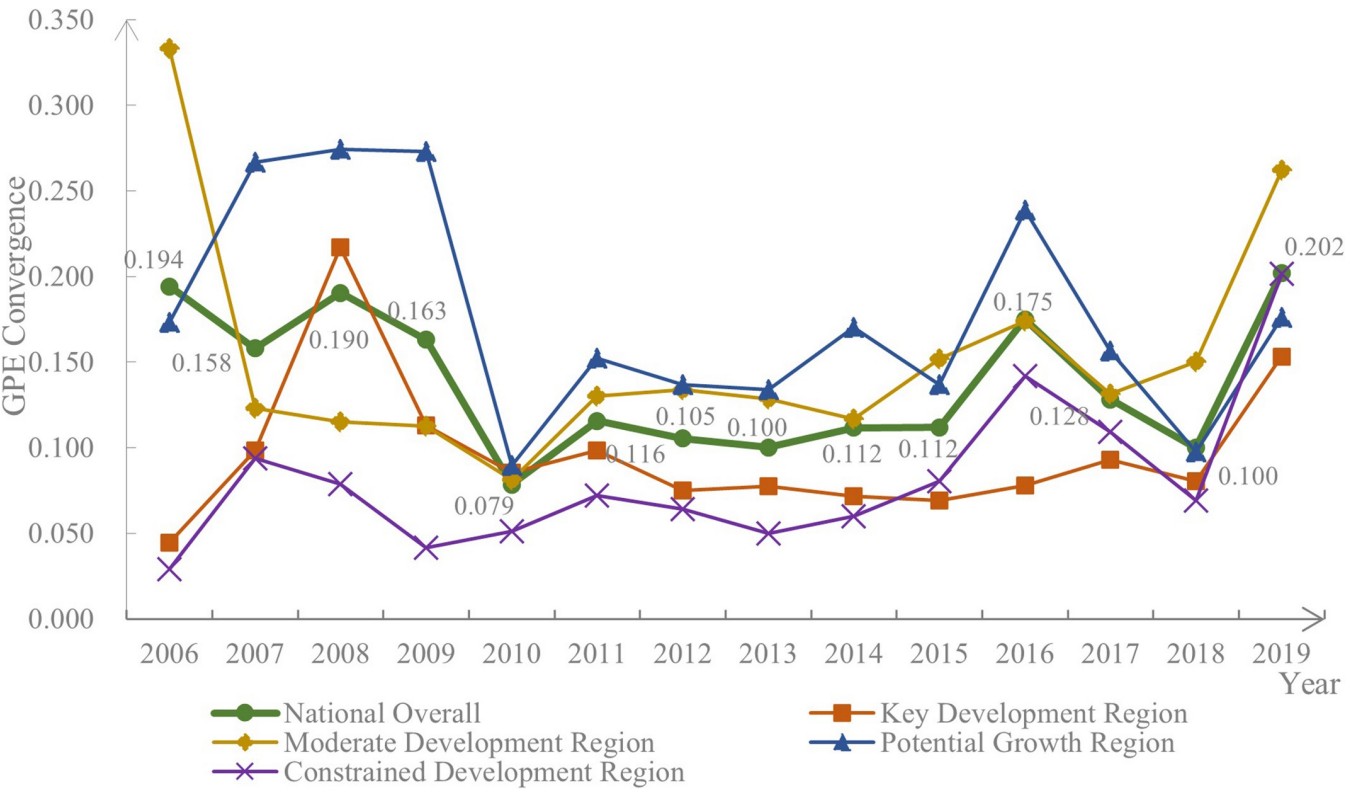

**Fig 3. Convergence trend of GPE in the hog breeding industry.**

severe impact on the hog breeding industry across all regions, leading to a decline in capacity and a reduction in the production gap, resulting in spatial divergence evolving into convergence. Although production capacity recovered after 2018, the direction and strength of adjustments made in the hog industry varied among provinces, which ultimately led to the expansion of spatial divergence.

## Driving factors of the spatial divergence

GPE can be broken down into the productivity of various inputs and outputs, and differences in regional inputs and outputs may impact the spatial divergence of GPE. Additionally, environmental investments, resource endowments, environmental load bearing, scale breeding, and production operation capacity may also have an effect on spatial divergence. This study employs factor detector and interaction detector techniques of Geodetector to examine the influence of constituent and external factors on the spatial-temporal divergence of GPE.

**Driving factors of the spatial divergence across periods.** Tables 5 and 6 present the outcomes of the factor detector and interaction detector analyses regarding the impact of input-output factors on the spatial divergence of GPE. The factor detector reveals that non-point source pollutants and labor inputs were the primary drivers of spatial divergence in GPE throughout the sample period. During the I-period, feed inputs and non-point source pollutants played significant roles in spatial divergence. In the II-period, spatial divergence in GPE primarily stemmed from labor inputs, carbon emissions, and non-point source pollutants. In the III-period, labor inputs and non-point source pollutants emerged as the main driving factors of spatial divergence.

**Table 5. Effect of input-output on spatial divergence of GPE across periods.**

| Periods | Labor input | Piglet input | Feed input | Water input | Energy input | Net Output | Carbon Emission | Pollutant emission |
|---------|-------------|--------------|------------|-------------|--------------|------------|-----------------|--------------------|
| Overall | 0.112 | 0.044 | 0.013 | 0.079 | 0.030 | 0.023 | 0.067 | 0.113 |
| I-period | 0.055 | 0.085 | 0.091 | 0.070 | 0.033 | 0.038 | 0.085 | 0.193 |
| II-period | 0.288 | 0.117 | 0.026 | 0.124 | 0.080 | 0.038 | 0.164 | 0.133 |
| III-period | 0.140 | 0.085 | 0.053 | 0.093 | 0.071 | 0.066 | 0.043 | 0.133 |

The interaction detector demonstrates that the combined effect of two factors on GPE exceeded that of individual factors. Among the interactions, the combination of labor inputs and non-point source pollutants had the most substantial impact on spatial divergence of GPE, with a value of 0.293. In the I-period, the interaction between feed inputs and non-point source pollutants had the greatest influence, reaching a value of 0.506. In the II-period, the interaction between piglet inputs and carbon emissions had the largest effect, reaching a value of 0.614. In the III-period, the interaction between piglet inputs and water input had the most significant impact, with a value of 0.529.

These findings indicate a decreasing influence of feed inputs and carbon emissions on the spatial divergence of GPE, while the impact of labor inputs, energy inputs, and net output on spatial divergence has gradually increased. The interactions between input factors consistently displayed a stronger effect, with the interaction between labor inputs and non-point source pollutants exhibiting a growing influence.

The impact of external factors, including environmental investment, resource endowment, environmental load bearing, scale breeding, and production operation capacity, on the GPE of the hog breeding industry is noteworthy. Tables 7 and 8 present the test results of the factor detector and interaction detector for each period. The factor detector indicates that resource endowment had the greatest influence on the spatial divergence of GPE, but its impact decreased over time, with q-statistic decreasing from 0.293 in the I-period to 0.153 in the III-period. This was followed by environmental load bearing, which saw a decrease in q-statistic from 0.091 to 0.076, and scale breeding, which experienced a decrease in q-statistic from 0.149 to 0.030. Meanwhile, the influence of environmental investment and production operation capacity on the spatial divergence decreased to 0.036 and 0.047, respectively, in the III-period. Resource endowment and environmental load bearing were the most important driving factors for the spatial divergence of GPE in the III-period, followed by production operation capacity, environmental investment, and scale breeding. The interaction detector shows that the interaction between all factors in each period had a greater impact than the effect of single factors. The interaction between resource endowment and environmental load bearing had the highest driving value of 0.388 across all periods. It can be concluded that resource

**Table 6. Interactive effect of input-output on spatial divergence of GPE across periods.**

| Periods | Leading interaction factor | q-statistic | Interaction |
|---------|----------------------------|-------------|-------------|
| Overall | Labor input ∩ Pollutant emission | 0.293 | Enhancement |
|  | Piglet input ∩ Carbon Emission | 0.292 | Enhancement |
| I-period | Feed input ∩ Pollutant emission | 0.506 | Enhancement |
|  | Piglet input ∩ Pollutant emission | 0.491 | Enhancement |
| II-period | Piglet input ∩ Carbon Emission | 0.614 | Enhancement |
|  | Labor input ∩ Pollutant emission | 0.588 | Enhancement |
| III-period | Piglet input ∩ Water input | 0.529 | Enhancement |
|  | Piglet input ∩ Net Output | 0.506 | Enhancement |

**Table 7. Effect of external factors on spatial divergence of GPE across periods.**

| Periods | Environmental Investment | Resource Endowment | Environmental Load Bearing | Scale Breeding | Production Operation |
|---------|--------------------------|--------------------|----------------------------|----------------|----------------------|
| Overall | 0.007 | 0.247 | 0.073 | 0.065 | 0.007 |
| I-period | 0.030 | 0.293 | 0.091 | 0.149 | 0.016 |
| II-period | 0.018 | 0.426 | 0.222 | 0.165 | 0.062 |
| III-period | 0.036 | 0.153 | 0.076 | 0.030 | 0.047 |

endowment and environmental load bearing were the most important factors for the spatial divergence of GPE among regions, leading to the expansion of the spatial divergence of GPE.

**Driving factors of the spatial divergence across regions.** Tables 9 and 10 exhibit the results of the factor detector and interaction detector for input-output on the spatial divergence across regions. The factor detector indicates that net output and non-point source pollutants were the primary driving factors for the spatial divergence of GPE in the key development region, with a q-statistic of 0.244 and 0.249, respectively. In the moderate development region, piglet input and net output had the greatest impact on spatial divergence, with q-statistics of 0.319 and 0.301, respectively. Meanwhile, piglet input had the most significant influence on the potential growth region, followed by water inputs, with q-statistics of 0.394 and 0.252, respectively. Carbon emissions and non-point source pollutants significantly affected the spatial divergence in the constrained development region, with q-values of 0.165 and 0.229, respectively.

The interaction detector suggests that the interaction of each factor played a major role in the spatial divergence of GPE. The interaction of net output and non-point source pollutants had the most significant effect on the spatial divergence in the key development region, with a value of 0.495. In the moderate development region, the interaction between labor input and net output was the most significant, with a value of 0.727. The largest effect of the interaction, with a q-statistic of 0.694, was between piglet input and net output in the potential growth region. Energy input and non-point source pollutants had the greatest impact on spatial divergence in the constrained development region, reaching 0.496.

Thus, it can be concluded that inputs were the primary reason for the spatial divergence of GPE in the hog breeding industry, while output status and pollution emission were the primary constraints for the green development of the hog breeding industry.

Tables 11 and 12 depict the effect of external factors on the spatial divergence of GPE in the hog breeding industry across regions. The factor detector reveals that environmental investment was the primary driver of spatial divergence in the key and moderate development regions, with a q-statistic of 0.256 and 0.145, respectively. Furthermore, resource endowment had a significant impact on the spatial divergence of the potential growth and constrained development regions, with q-statistics of 0.528 and 0.165, respectively. The interaction detector indicates that the interaction between environmental investment and environmental load bearing had the greatest impact on the spatial divergence in the key and moderate development regions, with q-statistics of 0.515 and 0.405, respectively. In the potential growth region,

**Table 8. Interactive effect of external factors on spatial divergence of GPE across periods.**

| Periods | Leading interaction factor | q-statistic | Interaction |
|---------|---------------------------|-------------|-------------|
| Overall | Resource Endowment ∩ Environmental Load Bearing | 0.388 | Enhancement |
| I-period | Resource Endowment ∩ Environmental Load Bearing | 0.528 | Enhancement |
| II-period | Resource Endowment ∩ Environmental Load Bearing | 0.575 | Enhancement |
| III-period | Resource Endowment ∩ Environmental Load Bearing | 0.410 | Enhancement |

Table 9. Effect of input-output on spatial divergence of GPE across regions.

| Regions | Labor input | Piglet input | Feed input | Water input | Energy input | Net Output | Carbon Emission | Pollutant emission |
|---|---|---|---|---|---|---|---|---|
| Key Development Region | 0.221 | 0.102 | 0.074 | 0.041 | 0.029 | 0.244 | 0.123 | 0.249 |
| Moderate Development Region | 0.178 | 0.319 | 0.127 | 0.077 | 0.225 | 0.301 | 0.112 | 0.204 |
| Potential Growth Region | 0.248 | 0.394 | 0.101 | 0.252 | 0.159 | 0.205 | 0.070 | 0.111 |
| Constrained Development Region | 0.098 | 0.124 | 0.045 | 0.148 | 0.060 | 0.133 | 0.165 | 0.229 |

the interaction of resource endowment and environmental load bearing had the most significant impact on spatial divergence, at 0.714. The spatial divergence in the constrained development region was mainly influenced by the interaction between scale breeding and resource endowment, with an effective degree of 0.382. It can be observed that environmental investment and environmental load bearing were the primary drivers of spatial divergence in the key and moderate development regions, while resource endowment had the most significant impact on spatial divergence in the potential growth and constrained development regions. The interaction between these factors formed each region's distinctive spatial divergence development pattern.

## Conclusions and suggestions

The study aimed to illuminate the spatial divergence of GPE within China's hog breeding industry and the underlying driving factors, offering valuable insights to facilitate the industry's pursuit of high-quality development. The GPE, along with its spatial divergence, was assessed utilizing the SBM model and divergence indexes across various regions from 2006 to 2019. Subsequently, the Geodetector was employed for further analysis of the driving forces behind spatial divergence. The findings revealed a positive trend in the GPE of the hog breeding industry in China and all regions, with an average value of 0.401 during the study period. Notably, key development areas exhibited the highest average GPE, while potential growth areas registered the lowest average GPE. Previous studies, such as Wang et al. [47] employed the SFA model to demonstrate the upward trend in cost efficiency for fattening pigs, sows, and piglets, with respective values of 0.77, 0.79, and 0.53. Wu et al. [48] utilized the SBM-Malmquist-Tobit model, revealing significant inter-provincial variations in the ecological efficiency of hog production, ranging from 0.557 to 1.19 across Chinese provinces in 2018. Hence, this paper aligns closely with the findings of prior research.

The drivers of GPE within China's hog breeding industry primarily encompass well-considered policy planning and technological advancements [46, 47]. To ensure an ample pork supply, diverse regions have devised hog development strategies based on resource allocation,

Table 10. Interactive effect of input-output on spatial divergence of GPE across regions.

| Regions | Leading interaction factor | q-statistic | Interaction |
|---|---|---|---|
| Key Development Region | Net Output ∩ Pollutant emission | 0.495 | Enhancement |
| | Water input ∩ Pollutant emission | 0.494 | Enhancement |
| Moderate Development Region | Labor input ∩ Net Output | 0.727 | Enhancement |
| | Labor input ∩ Energy input | 0.711 | Enhancement |
| Potential Growth Region | Piglet input ∩ Net Output | 0.694 | Enhancement |
| | Piglet input ∩ Feed input | 0.687 | Enhancement |
| Constrained Development Region | Energy input ∩ Pollutant emission | 0.496 | Enhancement |
| | Feed input ∩ Water input | 0.485 | Enhancement |

**Table 11. Effect of external factors on the spatial divergence of GPE across regions.**

| Regions | Environmental Investment | Resource Endowment | Environmental Load Bearing | Scale Breeding | Production Operation |
|---|---|---|---|---|---|
| Key Development Region | 0.256 | 0.094 | 0.138 | 0.150 | 0.106 |
| Moderate Development Region | 0.145 | 0.096 | 0.059 | 0.013 | 0.055 |
| Potential Growth Region | 0.123 | 0.528 | 0.260 | 0.134 | 0.131 |
| Constrained Development Region | 0.071 | 0.165 | 0.045 | 0.094 | 0.029 |

industrial foundations, environmental capacities, and developmental requirements. Under the auspices of the sustainable green development paradigm, GPE has exhibited continuous advancement. However, the industry faces risks associated with epidemics and market fluctuations, which impede its stability. The promotion of large-scale breeding practices and increased investments in environmental management bolster the progression of eco-friendly breeding technologies, thereby furnishing effective safeguards against breeding risks. Moreover, these endeavors facilitate the enhancement of input-output efficiency and GPE.

Regarding temporal progression, the spatial divergence of GPE within China's hog breeding industry undergoes a cycle of contraction followed by expansion, characterized by significant fluctuations and dispersion trends across the nation and its regions, thereby affirming the conclusions drawn in existing studies [47]. External risks, along with the industry's limited capacity to cope with such risks, profoundly influence the evolution of spatial divergence and contribute to fluctuations in GPE [49, 50]. Unexpected occurrences like outbreaks of diseases such as PRRS and ASF, as well as social incidents like the lean meat powder scandal, trigger unforeseen fluctuations in GPE and its spatial divergence, exerting adverse effects on the stable and sustainable development of the hog breeding industry. As the backbone of China's animal husbandry sector, the hog breeding industry's frequent market fluctuations have garnered widespread attention. Support for the industry's green development, encompassing capital, technology, and policy aspects, has spurred continuous growth in GPE amid these fluctuations. Furthermore, differing developmental foundations and strategic orientations across regions, as well as the formulation of distinct green development measures, inevitably contribute to the expansion of spatial divergence.

The spatial divergence of GPE witnessed expansion across all regions, excluding the key development region. The driving factors behind spatial divergence vary across different regions, predominantly influenced by resource endowment and environmental factors. While existing studies have extensively examined the influencing factors on GPE itself, exploring the causes of temporal fluctuations and the current state of GPE in various regions [46, 47, 49], limited attention has been devoted to the causes of spatial divergence, which this paper seeks to address. The findings reveal distinct driving factors contributing to GPE spatial divergence in different regions, primarily encompassing input-output factors and external influences. Among the external factors, environmental governance and environmental carrying capacity play significant roles as driving forces for GPE spatial divergence in key and moderate development areas, while resource endowment serves as the primary driving factor for spatial divergence in

**Table 12. Interaction effect of external factors on the spatial divergence of GPE across regions.**

| Regions | Leading interaction factor | q-statistic | Interaction |
|---|---|---|---|
| Key Development Region | Environmental Investment ∩ Environmental Load Bearing | 0.515 | Enhancement |
| Moderate Development Region | Environmental Investment ∩ Environmental Load Bearing | 0.405 | Enhancement |
| Potential Growth Region | Resource Endowment ∩ Environmental Load Bearing | 0.714 | Enhancement |
| Constrained Development Region | Resource Endowment ∩ Scale Breeding | 0.382 | Enhancement |

potential growth and restricted development areas. In terms of input-output factors, the impact of feed input and carbon emissions on spatial divergence gradually diminishes, while the influence of labor input, energy input, and net output on spatial divergence progressively increases. The disparities in various resource inputs emerge as the primary reasons for spatial divergence in GPE among regions, with output status and pollution emissions acting as crucial constraints on the green development of the hog breeding industry in each respective region.

In general, the spatial divergence of GPE in each region is primarily influenced by the regional endowment of resources and environmental factors, aligning with the conclusions drawn from previous research [12]. Hogs exhibit characteristics such as dense farming, significant emissions, and substantial governance challenges, while divergence in resources and the environment play a decisive role in determining the capacity and level of green development in hog breeding. Therefore, the future endeavor lies in addressing resource and environmental constraints, as well as bridging the gap in industrial development among regions through technological advancements and strategic planning.

To mitigate the impact of GPE's spatial divergence on industrial development, a series of targeted measures are imperative. Firstly, the green breeding technology level needs to be enhanced, with particular focus on the research and development of labor-saving technology, material-saving technology, and emission-reduction technology, to enable the automation, mechanization, cleanliness, and efficiency of the hog breeding industry. Secondly, the key development region should combine breeding and planting, with increased investment in environmental management, optimization of livestock breed structure, and the development of hog breeds adapted to regional characteristics. Thirdly, clean and standardized hog breeding actions should be taken in the moderate development region, with acceleration in the promotion of clean technologies and equipment, and standardized breeding for reduced pollutant emissions. Fourthly, large-scale hog breeding, following the principle of "land-based breeding", should be carried out in the potential growth region, with development of large-scale, specialized, intensive breeding, and cultivation of leading enterprises. Lastly, high-end boutique actions should be carried out in the constrained development region, to extend the industrial chain, deepen product processing, and improve breeding efficiency.

However, this paper has some limitations. Firstly, it did not examine the differences in GPE among different scales of hog breeding, but instead included it as a driving factor in the analysis system to investigate the impact of scaled breeding on spatial divergence. Therefore, this paper considers the hog breeding industry as a whole and explores the spatial divergence of GPE and its driving factors. Secondly, a variety of factors at the macro, meso, and micro levels all affect the spatial divergence of GPE in the hog breeding industry. This paper integrates previous research and identifies some under-explored yet crucial industry-level factors for analysis. Future studies could select variables from multiple levels and perspectives to construct a comprehensive analysis framework of driving factors. Furthermore, due to data availability and study period, the development status of the hog industry after the COVID-19 outbreak was not fully addressed, and the epidemic's impact on the industry requires further exploration to provide an empirical reference for the Chinese hog breeding industry to respond to external shocks.

## Supporting information

**S1 Data.**
(XLSX)

## Acknowledgments

We would like to thank all volunteers for participation in this study.

## Author Contributions

**Conceptualization:** Yifan Ji, Chun Li, Tao Xu.

**Data curation:** Yifan Ji, Ningjie Li, Chun Li.

**Formal analysis:** Yifan Ji, Zejun He.

**Funding acquisition:** Zejun He, Tao Xu.

**Methodology:** Yifan Ji.

**Project administration:** Zejun He, Tao Xu.

**Software:** Yifan Ji, Ningjie Li.

**Supervision:** Zejun He, Tao Xu.

**Writing – original draft:** Yifan Ji.

**Writing – review & editing:** Ningjie Li, Chun Li, Tao Xu.

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
