## [Decision Letter · Decision Letter 0]

29 Mar 2023

PONE-D-23-04011Green production efficiency of hog breeding industry in China: spatial stratified heterogeneity and its driving factorsPLOS ONE

Dear Dr. Xu,

Thank you for submitting your manuscript to PLOS ONE. After careful consideration, we feel that it has merit but does not fully meet PLOS ONE’s publication criteria as it currently stands. Therefore, we invite you to submit a revised version of the manuscript that addresses the points raised during the review process.

We look forward to receiving your revised manuscript.

Kind regards,

Chaohai Shen

Academic Editor

PLOS ONE

Journal Requirements:

"On behalf of all authors, disclose any competing interests that could be perceived to bias this work—acknowledging all financial support and any other relevant financial or non-financial competing interests."

Additional Editor Comments:

Dear Authors,

I have received the required number of reviewer reports. Based on their suggestions, together with my own review, I think the paper has some potential of attracting readers of the Journal. Therefore, I would like to invite you to revise the paper following comments from the reviewers (Reviewers 1, 2 and 3) and the comments from me. My comments are as follows.

1. You must rewrite the introduction part. Currently, your introduction is poorly organized. You must well motivate the topic with facts or data, as most of the well-written papers do. You must clearly propose your research questions based on real-world facts and latest literature so that the readers can easily figure out why your work is important. Particularly, you must clearly state your innovative findings. As you know, the topic you are working on is well investigated by many researchers, you must try your best to highlight your contributions.

2. You must update your literature. Currently, you missed many latest and important papers as highlighted by the reviewers. It is usually hard to convince readers that your findings are academically important without solid literature support. Accordingly, you must rewrite the literature review section.

3. You must clearly explain how you processed the data step by step. You must give the criteria for selecting or eliminating observations clearly. You must clearly state the number of observations left after each selection or elimination. You must carefully justify each selection or elimination, preferably with latest literature.

4. You must explain why you selected the variables for the empirical work by citing important papers.

5. You must provide some discussion on the robustness of your findings.

6. You must improve the writings of this paper with the help from native speakers or editing service. In addition, you need to format the paper following the guidance if necessary.

7. If you are willing to revise the paper, you must make sure the resubmitted manuscript is thoroughly proofread, so that there is no typo.

Reviewers' comments:

Reviewer's Responses to Questions

**Comments to the Author**

1. Is the manuscript technically sound, and do the data support the conclusions?

Reviewer #1: Yes

Reviewer #2: Yes

Reviewer #3: No

2. Has the statistical analysis been performed appropriately and rigorously? 

Reviewer #1: Yes

Reviewer #2: Yes

Reviewer #3: No

3. Have the authors made all data underlying the findings in their manuscript fully available?

Reviewer #1: No

Reviewer #2: Yes

Reviewer #3: No

4. Is the manuscript presented in an intelligible fashion and written in standard English?

Reviewer #1: No

Reviewer #2: Yes

Reviewer #3: No

5. Review Comments to the Author

Reviewer #1: 1. In your literature review, only the importance of pig breeding to China is reflected. I hope you can add a lot in the introduction about the proportion of pig breeding in other countries? Or how this result can be used for reference by other countries.

2. Why is the selection of the time stage based on China's five-year plan, and where is the specific impact on the pig breeding industry or the policy reflected? Because there are a large number of long data in the existing Chinese papers, which are divided into long time lines according to policies, such as the development of pig breeding industry since the reform and opening up.

3.172-174, should it be called the calculation of carbon equivalent? It is suggested that the calculation of carbon dioxide, methane and nitrous oxide should be uniformly multiplied by the global warming potential value, or the conversion coefficient of carbon equivalent should be uniformly used for carbon dioxide, methane and nitrous oxide. Of course, you can insist on your calculation method, and please give me an authoritative source of literature.

4.212-217 rows, a. Theil index, Gini coefficient and mean logarithmic dispersion can only be used as common statistical indicators to measure the difference. The problem of regional differences needs to be systematically analyzed by combining and using spatial econometric models, and pay attention to the correctness of the statements and methods in your article. B. Dagum Gini coefficient will also reflect regional differences and inter-regional differences better than the Gini coefficient you use. C. Confirm whether the source of your reference is correct.

5.Line 486, please supplement the limitations of your article in the discussion section, not just mention the innovation you think.

Reviewer #2: The purpose of this paper is to analyze the hog breeding industry's green production efficiency and its spatial stratified.

I list my comments as follows

1. Please define green production efficiency.

2. The maximum value of green production efficiency calculated by the author is 0.49, which seems to be inconsistent with the situation of pig breeding in China. Please mark clearly the basis for selecting the input and output indicators of green production efficiency, especially the basis for selecting the undesirable output, so as to judge whether the author has ignored the important input and output variables.

3. Please conduct descriptive statistical analysis of variables.

4. The author does not discuss the difference between different scale pig farming.

5. The research period ends in 2019. In 2018, the African swine fever caused a huge impact on pig farming in China, which needs further discussion by the author.

6. The author did not include labor resources, capital resources, GDP, infrastructure development level and other important influencing factors in the selection of driving factors. Please explain it and provide theoretical explanations for the selected variables.

7. Please note the source that the output value of the hog industry accounts for more than 50% of China's gross domestic product of animal" husbandry and supplies more than 56% of all meat consumption, which has a pivotal position in China.

Reviewer #3: The author's research does not have any innovation. This topic has been studied by many people, so I hold a questioning attitude towards this topic.

1. The research topic has already been studied by others. Is not an innovative theme. Please explain the difference between you and Zhong et al (2022).

2. Language needs further improvement

6. PLOS authors have the option to publish the peer review history of their article (what does this mean?). If published, this will include your full peer review and any attached files.

Reviewer #1: No

Reviewer #2: No

Reviewer #3: No

---

## [Author Response · Author response to Decision Letter 0]

15 May 2023

Response to Reviewers

Dear editors and reviewers: 

Thank you very much for your letter and the reviewers' comments on our paper “Green production efficiency of China’s hog breeding industry: spatial divergence and its driving factors” (Manuscript ID: PONE-D-23-04011).

We have learned much from the reviewers’ comments, which are fair, encouraging, and constructive. After carefully studying the comments and your advice, we have made a corresponding revision, and the response to the comments is enclosed at the end of this letter, with the revision contents in this manuscript marked in red. If you have any questions about this paper, please don’t hesitate to contact us.

Editor Comments

[Comment 1] You must rewrite the introduction part. Currently, your introduction is poorly organized. You must well motivate the topic with facts or data, as most of the well-written papers do. You must clearly propose your research questions based on real-world facts and latest literature so that the readers can easily figure out why your work is important. Particularly, you must clearly state your innovative findings. As you know, the topic you are working on is well investigated by many researchers, you must try your best to highlight your contributions. 

[Response] We have rewritten the introduction. The research topic has been stimulated from both practical and theoretical issues, and the corresponding supporting data and literature have been provided (On Page 3~6). And based on the literature analysis, the importance of our work and possible innovative findings are presented in the last paragraph of the introduction “Firstly, it concentrates on the spatial divergence of GPE in the hog breeding industry and scrutinizes the sources of its spatial divergence, providing a reference basis for the industry's green and coordinated development. Secondly, it examines the GPE characteristics in different areas based on the division of functional areas of the hog breeding industry to enhance the precision and relevance of research outcomes. Thirdly, the paper analyses the driving factors of spatial divergence of GPE in hog breeding industry from both internal and external perspectives, exploring the causes of its spatial divergence. The study results can serve as a reference for promoting coordinated regional development of the hog breeding industry, and thus improving the quality of its development”. (On Page 6) 

[Comment 2] You must update your literature. Currently you missed many latest and important papers as highlighted by the reviewers. It is usually hard to convince readers that your findings are academically important without solid literature support. Accordingly, you must rewrite the literature review section.

[Response] We have supplemented and updated the literature. And the literature review section has been integrated with the introduction to highlight the importance and innovative contributions of our research topic and to provide solid literature support for the formulation of the problem. (On Page 4~5)

[Comment 3] You must clearly explain how you processed the data step by step. You must give the criteria for selecting or eliminating observations clearly. You must clearly state the number of observations left after each selection or elimination. You must carefully justify each selection or elimination, preferably with latest literature. 

[Response] We have reintroduced a detailed description of the calculation process of each input-output indicator of GPE, improved the data sources and data processing process (Shown in formula 1~9 on Page 8~10). We have also supplemented the basis for the selection of input-output indicators by selecting the most recent literature as far as possible to provide a more adequate justification for the selection and exclusion of each indicator. (On Page 7~8, reference 24~29)

[Comment 4] You must explain why you selected the variables for the empirical work by citing important papers. 

[Response] We have supplemented the evidence for the selection of the driver indicators by citing key literature and explaining the rationale for the selection. (On Page 11~13, reference 32~38) 

[Comment 5] You must provide some discussion on the robustness of your findings.

[Response] We have argued for the robustness of our research work by comparing our findings with those of the important literature. “Wang et al. (2021) used the SFA model to demonstrate that the cost efficiency of fattening pigs, sows, and piglets in China was 0.77, 0.79, and 0.53, respectively, with an upward trend development of cost efficiency (49). Wu et al. (2022) used the SBM-Malmquist-Tobit model to demonstrate that the ecological efficiency of hog production in Chinese provinces ranged from 0.557 to 1.19 in 2018, indicating significant inter-provincial differences. Thus, this paper is highly consistent with previous studies' findings (50)”. (On Page 36)

[Comment 6] You must improve the writings of this paper with the help from native speakers or editing service. In addition, you need to format the paper following the guidance if necessary. 

[Response] We have re-edited and refined the language of the paper and revised the format of the manuscript according to journal guidelines.

[Comment 7] If you are willing to revise the paper, you must make sure the resubmitted manuscript is thoroughly proofread, so that there is no typo. 

[Response] We have thoroughly proofread the manuscript to ensure that there are no typos.

Reviewer 1:

[Comment 1] In your literature review, only the importance of pig breeding to China is reflected. l hope you can add a lot in the introduction about the proportion of pig breeding in other countries? Or how this result can be used for reference by other countries. 

[Response] We have added the importance of hog breeding to the world in the introduction as well as in the literature review, providing support from both data and literature, including the proportion of hogs raised in the world and the contribution of pork. “Pork production accounts for 32.57% of the total meat output globally, with 100.853 million tons of pork consumed worldwide in 2021, serving as a significant protein source for human consumption”, “China, as the largest pork consumer and producer globally, where pork consumption accounts for 44% of the world's total consumption and production accounts for 38.3% of the total global production”. (On Page 3, reference 1~2)

[Comment 2] Why is the selection of the time stage based on China's five-year plan, and where is the specific impact on the pig breeding industry or the policy reflected? Because there are a large number of long data in the existing Chinese papers, which are divided into longtime lines according to policies, such as the development of pig breeding industry since the reform and opening up. 

[Response] We provide an adequate industrial policy basis for the existing period division, “Classification basis: the Eleventh Five-Year Plan for the Development of National Animal Husbandry (2006-2010), the Twelfth Five-Year Plan for the Development of National Animal Husbandry (2011-2015) and the National Pig Production Development Plan (2016-2020) issued by the Ministry of Agriculture of China in 2006, 2011 and 2016, respectively, are long-term plans, which provide for the development of each five-year. The plans lay out the development path of the industry and provide goals and directions for the development vision of the hog breeding industry, which have an important guiding role for the industry. Examining the GPE of the hog breeding industry in different planning periods can reflect the development characteristics of the industry led by policies, so this paper divides the sample period into I-period (2006-2010), II-period (2011-2015) and III-period (2016-2019).” (On Page 46, endnote 5). The livestock policies introduced by the Chinese government every five years have an important impact on the hog breeding industry, and dividing the sample period into short five-year periods helps to further explore the industrial development trend and the impact of government policies on the green development of the hog breeding industry.

[Comment 3] 172-174, should it be called the calculation of carbon equivalent? It is suggested that the calculation of carbon dioxide, methane and nitrous oxide should be uniformly multiplied by the global warming potential value, or the conversion coefficient of carbon equivalent should be uniformly used for carbon dioxide, methane and nitrous oxide. Of course, you can insist on your calculation method, and please give me an authoritative source of literature. 

[Response] We have reorganized the process of measuring undesirable output indicators and changed them to an equation form, using carbon equivalent conversion factors for C, CH4 and N2O consistently. (Shown in formula 1~9 on Page 8~10)

[Comment 4] 212-217 rows, a. Theil index, Gini coefficient and mean logarithmic dispersion can only be used as common statistical indicators to measure the difference. The problem of regional differences needs to be systematically analyzed by combining and using spatial econometric models, and pay attention to the correctness of the statements and methods in your article. B.Dagum Gini coefficient will also reflect regional differences and inter-regional differences better than the Gini coefficient you use. C. Confirm whether the source of your reference is correct. 

[Response] A. We have checked and revised the statements of words and phrases in the paper to reflect the level of variation of GPE in hog breeding industry through statistical indicators. (On Page 15~16). B. The reason we use the Gini coefficient is that it is complementary to the Thiel index and the mean log deviation. The above three methods are sensitive to changes in high, medium, and low levels, respectively, so scholars mostly analyze regional divergence in development levels based on the comparison results of the three methods. C. We have updated and checked references to ensure the accuracy of the sources. (Reference 39~43)

[Comment 5] Line 486, please supplement the limitations of your article in the discussion section, not just mention the innovation you think. 

[Response] We have added the limitations of this paper in the discussion section. “However, this paper has some limitations. Firstly, it did not examine the differences in GPE among different scales of hog breeding, but instead included it as a driving factor in the analysis system to investigate the impact of scaled breeding on spatial divergence. Therefore, this paper considers the hog breeding industry as a whole and explores the spatial divergence of GPE and its driving factors. Secondly, a variety of factors at the macro, meso, and micro levels all affect the spatial divergence of GPE in the hog breeding industry. This paper integrates previous research and identifies some under-explored yet crucial industry-level factors for analysis. Future studies could select variables from multiple levels and perspectives to construct a comprehensive analysis framework of driving factors. Furthermore, due to data availability and study period, the development status of the hog industry after the COVID-19 outbreak was not fully addressed, and the epidemic's impact on the industry requires further exploration to provide an empirical reference for the Chinese hog breeding industry to respond to external shocks.” (On Page 38)

Reviewer 2:

[Comment 1] Please define green production efficiency.

[Response] We have defined the concept of green production efficiency based on existing literature. “GPE is a study of the degree of coordination between economic development and environmental resources, based on the limited environmental carrying capacity, to achieve the goal of energy conservation, consumption reduction, and pollution reduction in the production process, using advanced management methods and technology. It not only evaluates the efficiency of resource inputs, such as land, labor, and capital but also takes environmental factors into account to comprehensively measure the overall efficiency of industries under the constraints of resource and environmental factors (4)”. (On Page 4) 

[Comment 2] The maximum value of green production efficiency calculated by the author is 0.49which seems to be inconsistent with the situation of pig breeding in China. Please mark clearly the basis for selecting the input and output indicators of green production efficiency, especially the basis for selecting the undesirable output, so as to judge whether the author has ignored the important input and output variables. 

[Response] We have explained the basis for the selection of green production efficiency input-output indicators, supplemented the literature for argumentation (On Page 7~10). In particular, the basis for the selection of undesirable output indicators and the calculation process are fully explained. (Shown in formula 1~9 on Page 8~10)

[Comment 3] Please conduct descriptive statistical analysis of variables.

[Response] We have performed descriptive statistical analysis of the variables, as shown in Table 3. (On page 18~19) 

[Comment 4] The author does not discuss the difference between different scale pig farming.

[Response] This paper does not examine the differences in GPE among different scales of hog breeding, but instead included it as a driving factor in the analysis system to investigate the impact of scaled breeding on spatial divergence. The drivers of GPE and its spatial divergence in different scales of hog farming are quite distinct, and it is difficult to be included in a framework for research. Therefore, this paper considers the hog breeding industry as a whole and explores the spatial divergence of GPE and its driving factors.

[Comment 5] The research period ends in 2019.In 2018, the African swine fever caused a huge impact on pig farming in China, which needs further discussion by the author.

[Response] Limited by data availability, our work only explores the green production efficiency of China's hog breeding industry from 2006 to 2019, and the impact of African swine fever on the industry in 2018 is reflected in the anomaly of the 2019 data, which is explained in the paper. “This was primarily due to the impact of African Swine Fever (ASF) in 2018, which significantly reduced hog breeding efficiency and spurred the rapid reorganization of the industry, culminating in the enhanced green development of the hog breeding industry and a corresponding surge in hog GPE in 2019” (On page 23). “In 2018, the outbreak of African Swine Fever (ASF) had a severe impact on the hog breeding industry across all regions, leading to a decline in capacity and a reduction in the production gap, resulting in spatial divergence evolving into convergence. Although production capacity recovered after 2018, the direction and strength of adjustments made in the hog industry varied among provinces, which ultimately led to the expansion of spatial divergence.” (On page 27) 

[Comment 6] The author did not include labor resources capital resources, GDP, infrastructure development level and other important influencing factors in the selection of driving factors. Please explain it and provide theoretical explanations for the selected variables. 

[Response] Variables such as labor resources, capital resources, and human resources have been included in the GPE measurement process, and exploring their effects on GPE may result in inaccurate results. Driving factors such as GDP and infrastructure development level have now been fully argued by scholars, so we have selected some other important factors to carry out the discussion. In addition, we have provided theoretical explanations for the selected variables. (On page 11~13)

[Comment 7] Please note the source that the output value of the hog industry accounts for more than 50% of China's gross domestic product of animal" husbandry and supplies more than56% of all meat consumption, which has a pivotal position in China.

[Response] We have marked the data sources in the article in the endnotes. (On page 46)

Reviewer 3:

[Comment 1] The research topic has already been studied by others. Is not an innovative theme. Please explain the difference between you and Zhong et al (2022).

[Response] The main difference between this paper and Zhong et al (2022) is that, firstly, this paper explores the factors driving the spatial divergence of GPE from the perspective of both internal and external factors, and examines the specific influence of each input-output indicator on GPE, which also further provides a basis for the selection of indicators and argues the reliability of the indicators. Secondly, this paper focuses on the driving factors of GPE spatial divergence instead of GPE itself, which is the main focus of previous studies, and explores the reasons for the formation of GPE spatial divergence to provide reference for reducing regional differences.

[Comment 2] Language needs further improvement.

[Response] We have re-corrected and re-condensed the language.

---

## [Decision Letter · Decision Letter 1]

6 Jun 2023

PONE-D-23-04011R1Green production efficiency of China's hog breeding industry: spatial divergence and its driving factorsPLOS ONE

Dear Dr. 徐,

Thank you for submitting your manuscript to PLOS ONE. After careful consideration, we feel that it has merit but does not fully meet PLOS ONE’s publication criteria as it currently stands. Therefore, we invite you to submit a revised version of the manuscript that addresses the points raised during the review process.

We look forward to receiving your revised manuscript.

Kind regards,

Chaohai Shen

Academic Editor

PLOS ONE

Journal Requirements:

Additional Editor Comments:

Dear Authors,

Thanks for submitting your revised manuscript. Two of the reviewers still have some concerns regarding your work. Please strictly follow their comments to make revisions.

Sincerely,

Reviewers' comments:

Reviewer's Responses to Questions

**Comments to the Author**

1. If the authors have adequately addressed your comments raised in a previous round of review and you feel that this manuscript is now acceptable for publication, you may indicate that here to bypass the “Comments to the Author” section, enter your conflict of interest statement in the “Confidential to Editor” section, and submit your "Accept" recommendation.

Reviewer #1: (No Response)

Reviewer #2: All comments have been addressed

Reviewer #3: All comments have been addressed

2. Is the manuscript technically sound, and do the data support the conclusions?

Reviewer #1: (No Response)

Reviewer #2: Yes

Reviewer #3: Yes

3. Has the statistical analysis been performed appropriately and rigorously? 

Reviewer #1: (No Response)

Reviewer #2: Yes

Reviewer #3: Yes

4. Have the authors made all data underlying the findings in their manuscript fully available?

Reviewer #1: (No Response)

Reviewer #2: Yes

Reviewer #3: Yes

5. Is the manuscript presented in an intelligible fashion and written in standard English?

Reviewer #1: (No Response)

Reviewer #2: Yes

Reviewer #3: Yes

6. Review Comments to the Author

Reviewer #1: Review comments

The author has indeed made significant improvements, but there are still some minor issues that need to be addressed. Here are the suggested modifications:

1.Please verify if the data for Labor input, Piglet input, and Feed input in Table 2 and Table 3 are consistent.

2.It is recommended to use the unit of US dollars for variables like Energy input and Water input throughout the entire text. Please make the necessary modifications.

3.It is advised to revise the format of Table 4. The Table currently appears unappealing and is somewhat difficult to comprehend. Please make it more visually appealing and easier to understand.

Reviewer #2: This paper analyzed the green production efficiency (GPE) and its spatial stratification heterogeneity in pig breeding industry, which provided the basis for rational layout of pig breeding industry and promoting the development of high quality industry. The GPE of pig breeding industry in 29 provinces (municipalities and districts) of China from 2006 to 2019 was estimated, and the spatial stratified heterogeneity of GPE and its driving factors were analyzed using spatial index and geographic detector, respectively. The paper conforms to the analysis paradigm of industrial economics, and the research content is in line with the direction of the journal. It can promote the high-quality development of the live pig industry, and has reference value for the government to formulate policies to cope with the corresponding economic development, so as to promote the green and healthy development of the live pig industry.

To sum up:

In the topic selection, the full text has a strong practical significance, can formulate the pig industry and other related policies.

In terms of research norms, it meets the requirements of scientific and rigorous academic papers, and the logical correlation between each chapter is strong.

In terms of theoretical value, it can provide real and practical data and promote the generation of new knowledge.

Although the two aspects of literature and writing standards still need to be strengthened, but the defects still cannot obscure the manuscript. It is therefore recommended that it be published with appropriate revisions.

The contents that still need to be modified are as follows:

1. It is best to avoid the phenomenon of title to title. After each title, there should be a general paragraph to summarize the main content and ideas of this chapter, and explain the relationship with the previous paragraph, so as to play the role of connecting the preceding and the following.

2. The discussion section at the end of the article is placed in the front as the background, which is not suitable for the end.

3. Conclusions and suggestions. Empirical results need to be further analyzed in combination with existing studies. This paper supplements the views of existing scholars in those aspects. Have any studies come to similar conclusions? If so, what does the paper add to the existing conclusions? It is suggested to combine the existing literature for in-depth analysis and discussion, not only stay in the experimental results, but also further practicallars.

Reviewer #3: (No Response)

7. PLOS authors have the option to publish the peer review history of their article (what does this mean?). If published, this will include your full peer review and any attached files.

Reviewer #1: No

Reviewer #2: No

Reviewer #3: No

---

## [Author Response · Author response to Decision Letter 1]

13 Jun 2023

Dear editors and reviewers: 

Thank you very much for your letter and the reviewers' comments on our paper “Green production efficiency of China’s hog breeding industry: spatial divergence and its driving factors” (Manuscript ID: PONE-D-23-04011).

We have learned much from the reviewers’ comments, which are fair, encouraging, and constructive. After carefully studying the comments and your advice, we have made a corresponding revision, and the response to the comments is enclosed at the end of this letter, with the revision contents in this manuscript marked in red. If you have any questions about this paper, please don’t hesitate to contact us.

Reviewer 1:

The author has indeed made significant improvements, but there are still some minor issues that need to be addressed. Here are the suggested modifications:

[Comment 1] Please verify if the data for Labor input, Piglet input, and Feed input in Table 2 and Table 3 are consistent.

[Response] We have conducted a thorough examination of Tables 2 and 3 to ensure the consistency of data concerning Labor input, Piglet input, and Feed input. (On Page 13 and 19)

[Comment 2] It is recommended to use the unit of US dollars for variables like Energy input and Water input throughout the entire text. Please make the necessary modifications.

[Response] We have applied the current year's exchange rate to convert all amounts from RMB to USD across the entirety of the text. (On Page 11, 13 and 19, in Tables 1, 2, and 3)

[Comment 3] It is advised to revise the format of Table 4. The Table currently appears unappealing and is somewhat difficult to comprehend. Please make it more visually appealing and easier to understand.

[Response] To enhance clarity and visual appeal, we have replaced Table 4 with Fig 1, presenting the results in a more comprehensible and captivating manner. (On Page 20)

Reviewer 2:

Although the two aspects of literature and writing standards still need to be strengthened, but the defects still cannot obscure the manuscript. It is therefore recommended that it be published with appropriate revisions.

The contents that still need to be modified are as follows:

[Comment 1] It is best to avoid the phenomenon of title to title. After each title, there should be a general paragraph to summarize the main content and ideas of this chapter, and explain the relationship with the previous paragraph, so as to play the role of connecting the preceding and the following.

[Response] First, we have scrutinized the entire text, rectifying instances where titles were used instead of headings. Second, we have appended a summary paragraph beneath each heading, succinctly capturing the main content and ideas of the chapter, while establishing connections with the preceding chapter's material.

“This chapter presents a comprehensive account of the methods and data sources employed in the research process of this paper. It elucidates the procedure for choosing indicators to measure GPE in the hog breeding industry, along with the underlying rationale for selecting the drivers. This chapter furnishes methods and material backing for calculating and analyzing the research findings in the subsequent section.” (In Lines 124~128)

“According to the arrangement of the study, this section selects the input-output variables required to measure GPE and the driving factors variables that may affect the spatial divergence of GPE to provide material support for the study.” (In Lines 150~152)

“This section provides a detailed description of the methods to be applied in this study, including the SBM model, the divergence index, and the Geodetector, which helps to improve the applicability of the study.” (In Lines 237~239)

“This chapter employs the aforementioned materials and methods to gauge the GPE of the hog breeding industry and conducts an analysis of the outcomes. It expounds upon the development characteristics and evolutionary trends of GPE in diverse regions and provinces across various time periods, examining both spatial and temporal dimensions, while offering profound interpretations for the underlying reasons. The findings of this chapter serve as the foundation for scrutinizing the spatial divergence of GPE and its driving factors.” (In Lines 310~315)

“Building upon the aforementioned measurement results, this section proceeds to delve into an in-depth analysis of the extent of divergence in GPE within various regions and provinces throughout the sample period. Expanding on this foundation, the dispersion trend of GPE divergence among different regions is explored through the implementation of an α convergence test, while elucidating the underlying reasons behind this evolving trend. Moreover, this chapter examines the distinctive features of spatial divergence in GPE within the hog breeding sector across different periods and regions, affirming the driving factors through the scrutiny of input-output factors and external factors.” (In Lines 384~391)

[Comment 2] The discussion section at the end of the article is placed in the front as the background, which is not suitable for the end.

[Response] We have integrated the background information discussed in the Discussion section into the Introduction, eliminating redundant content. “It not only evaluates the efficiency of resource inputs, such as land, labor, and capital but also takes environmental factors into account to comprehensively measure the overall efficiency of industries under the constraints of resource and environmental factors” (In Lines 67~70)

“This spatial divergence not only leads to an unbalanced allocation of resources and technologies and a loss of coordination in industrial development, but also affects the effectiveness of emission reduction and environmental management, resulting in social green inequity and hindering high-quality development in the industry” (In Lines 91~94)

[Comment 3] Conclusions and suggestions. Empirical results need to be further analyzed in combination with existing studies. This paper supplements the views of existing scholars in those aspects. Have any studies come to similar conclusions? If so, what does the paper add to the existing conclusions? It is suggested to combine the existing literature for in-depth analysis and discussion, not only stay in the experimental results, but also further practicallars.

[Response] First, we have amalgamated the "Discussion" and "Conclusion" sections into the comprehensive "Conclusions and suggestions" section.

Second, we have meticulously restructured this section, delving deeply into the empirical findings and providing thorough explanations for the observed outcomes. (In Lines 543~605)

Third, we have conducted a meticulous analysis and discussion of the results within the framework of existing literature, amplifying the credibility of this paper's findings. (In Lines 543~605)

Fourth, we have outlined the novel additions and contributions of this study to existing research, thereby enhancing the practical significance of the obtained results. “While existing studies have extensively examined the influencing factors on GPE itself, exploring the causes of temporal fluctuations and the current state of GPE in various regions [45, 46, 48], limited attention has been devoted to the causes of spatial divergence, which this paper seeks to address.” (In Lines 583~586)

Reviewer 3:

(No Response)

Reference revision

We removed the original [45]-[51] from the list of references and added the existing [45]-[48].

---

## [Decision Letter · Decision Letter 2]

20 Jun 2023

Green production efficiency of China's hog breeding industry: spatial divergence and its driving factors

PONE-D-23-04011R2

Dear Dr. 徐,

We’re pleased to inform you that your manuscript has been judged scientifically suitable for publication and will be formally accepted for publication once it meets all outstanding technical requirements.

Kind regards,

Chaohai Shen

Academic Editor

PLOS ONE

Additional Editor Comments (optional):

Reviewers' comments:

Reviewer's Responses to Questions

**Comments to the Author**

1. If the authors have adequately addressed your comments raised in a previous round of review and you feel that this manuscript is now acceptable for publication, you may indicate that here to bypass the “Comments to the Author” section, enter your conflict of interest statement in the “Confidential to Editor” section, and submit your "Accept" recommendation.

Reviewer #1: All comments have been addressed

2. Is the manuscript technically sound, and do the data support the conclusions?

Reviewer #1: Yes

3. Has the statistical analysis been performed appropriately and rigorously? 

Reviewer #1: Yes

4. Have the authors made all data underlying the findings in their manuscript fully available?

Reviewer #1: Yes

5. Is the manuscript presented in an intelligible fashion and written in standard English?

Reviewer #1: Yes

6. Review Comments to the Author

Reviewer #1: (No Response)

7. PLOS authors have the option to publish the peer review history of their article (what does this mean?). If published, this will include your full peer review and any attached files.

Reviewer #1: No

---

## [Editor Report · Acceptance letter]

26 Jun 2023

PONE-D-23-04011R2 

Green production efficiency of China's hog breeding industry: spatial divergence and its driving factors 

Dear Dr. Xu:

I'm pleased to inform you that your manuscript has been deemed suitable for publication in PLOS ONE. Congratulations! Your manuscript is now with our production department. 

Kind regards, 

on behalf of

Dr. Chaohai Shen 

Academic Editor

PLOS ONE